# Acute effect of low-load resistance exercise with blood flow restriction on oxidative stress biomarkers: A systematic review and meta-analysis

**João Vitor Ferlito**[1]☯*, **Nicholas Rolnick**[2]☯, **Marcos Vinicius Ferlito**[1], **Thiago De Marchi**[3], **Rafael Deminice**[4], **Mirian Salvador**[1]

**1** Oxidative Stress and Antioxidant Laboratory, Postgraduate Program in Biotechnology, University of Caxias Do Sul, Caxias do Sul, Brazil, **2** The Human Performance Mechanic, Lehman College, New York, NY, United States of America, **3** Laboratory of Phototherapy and Innovative Technologies in Health (LaPIT), Postgraduate Program in Rehabilitation Sciences, Nove de Julho University (UNINOVE), São Paulo, SP, Brazil, **4** Department of Physical Education, State University of Londrina, Londrina, Brazil

☯ These authors contributed equally to this work.
* joaoferlito@yahoo.com.br

## Abstract

### Background

The purpose of this review was to analyze the acute effects of low-load resistance exercise with blood flow restriction (LLE-BFR) on oxidative stress markers in healthy individuals in comparison with LLE or high-load resistance exercise (HLRE) without BFR.

### Materials and methods

A systematic review was performed in accordance with the PRISMA (Preferred Reporting Items for Systematic Reviews and Meta-Analyses) guidelines. These searches were performed in CENTRAL, SPORTDiscus, EMBASE, PubMed, CINAHL and Virtual Health Library- VHL, which includes Lilacs, Medline and SciELO. The risk of bias and quality of evidence were assessed through the PEDro scale and GRADE system, respectively.

### Results

Thirteen randomized clinical trials were included in this review (total n = 158 subjects). Results showed lower post-exercise damage to lipids (SMD = -0.95 CI 95%: -1.49 to -0. 40, $I^2$ = 0%, p = 0.0007), proteins (SMD = -1.39 CI 95%: -2.11 to -0.68, $I^2$ = 51%, p = 0.0001) and redox imbalance (SMD = -0.96 CI 95%: -1.65 to -0.28, $I^2$ = 0%, p = 0.006) in favor of LLRE-BFR compared to HLRE. HLRE presents higher post-exercise superoxide dismutase activity but in the other biomarkers and time points, no significant differences between conditions were observed. For LLRE-BFR and LLRE, we found no difference between the comparisons performed at any time point.

**Data Availability Statement:** All relevant data are within the manuscript and its Supporting Information files.

**Funding:** The authors received no specific funding for this work.

**Competing interests:** NR is the founder of THE BFR PROS, a BFR education company that provides BFR training workshops to fitness and rehabilitation professionals across the world using a variety of BFR devices. NR has no financial relationships with any cuff manufacturers/distributors. The remaining authors declare that they have no conflict of interests. This does not alter our adherence to all PLOS ONE policies on sharing data and materials.

## Conclusions

Based on the available evidence from randomized trials, providing very low or low certainty of evidence, this review demonstrates that LLRE-BFR promotes less oxidative stress when compared to HLRE but no difference in levels of oxidative damage biomarkers and endogenous antioxidants between LLRE.

## Trial registration

Register number: PROSPERO number: CRD42020183204.

## Introduction

Recent evidence shows that low-load resistance exercise (LLRE, 20% to 50% of one repetition maximum-1RM) with blood flow restriction (LLRE-BFR)—also called the Kaatsu method [1]—induces muscle adaptations similar to high-load resistance exercise (HLRE) in clinical musculoskeletal rehabilitation [2], elderly [3] and athlete [4] populations. LLRE-BFR is performed with the application of external pressure through a pressurized cuff or elastic strap applied over the proximal third of the upper or lower limbs [1]. Studies suggest that the externally applied pressure reduces arterial flow, and occludes venous blood flow distal to the site of application, causing a decrease of blood entry into the muscle [5,6]. The reduction of blood flow creates an ischemic/hypoxic environment in the muscle, increasing metabolic stress, one of the proposed mechanisms thought to induce muscle strength and hypertrophy [5,7,8]. Pearson and Hussain [8] hypothesize that metabolic stress associated with LLRE-BFR promotes muscle growth through earlier fast-twitch muscle fiber recruitment, greater hormonal production, cellular swelling, and production of reactive oxygen species (ROS).

Generation of ROS is directly related with several physiological processes in skeletal muscle, including the control of gene expression, regulation of cell signaling pathways, and modulation of skeletal muscle force production [9,10]. Although excessive ROS production may cause muscle damage and strength loss, low to moderate concentrations of ROS following resistance exercise may act as signaling molecules for muscle adaptation [11]. Furthermore, post resistance exercise, physiological responses such as the infiltration of phagocytes (i.e., neutrophils and macrophages) at the site of injury are necessary for neuromuscular adaptation [8]. This exercise-induced inflammatory response also contributes to increase of ROS and oxidative damage to biomolecules [9].

One of the mechanisms for increasing production of exercise-induced ROS occurs through ischemia-reperfusion [12]. Ischemia-reperfusion can be summarized as the deprivation of blood flow and lack of oxygen (ischemia/hypoxia), followed by the restoration of blood flow and supercompensation of muscle tissue oxygenation (reperfusion) [13]. During exercise performance between the concentric and eccentric muscle action stages, the muscle fiber may experience temporary ischemia from exercise demand and microvasculature compression [14]. After completion of the exercise bout, perfusion is re-established as the local microvasculature is no longer compressed through muscular contraction. This may induce activation of xanthine oxidase, NADPH oxidase, and the leakage of electrons by the mitochondrial electron transport chain, increasing ROS production in skeletal muscle [12].

Indeed, studies have demonstrated LLRE-BFR induces greater intramuscular deoxygenation compared to LLRE and HLRE without BFR [15,16]. Thus, the hypoxic environment and

local cellular $PO_2$ reductions generated during LLRE-BFR could cause a reduced state of the mitochondrial electron transport chain, producing greater mitochondrial ROS emissions. The increase in ROS production may be triggered by reperfusion of ischemic tissue, likely initiating oxidative stress from elevations in local muscular oxygen consumption [17]. In this way, it is hypothesized that ischemia-reperfusion potentiated during skeletal muscle contraction with BFR may be a mechanism promoting ROS production and subsequent muscle adaptations [18], given it can promote comparable adaptations to HLRE despite lower exercise loads. Therefore, it is unclear whether resistance exercise with BFR generates similar or greater oxidative stress markers than resistance exercise without BFR. Understanding the effects of BFR on oxidative stress is central to understanding the cellular mechanisms involved in the muscular adaptations following longitudinal BFR resistance exercise programs. This is relevant considering the growing body of literature supporting the use of BFR to induce positive musculoskeletal benefits comparable to HLRE and superior to LLRE [6].

The aim of this review was to systematically analyze the evidence on exercise-induced oxidative stress in resistance exercise with and without BFR. The research questions for this systematic review were: (1) Does a LLRE-BFR session generate antioxidant responses and oxidative stress greater or similar to HLRE? and (2) Does a session of LLRE-BFR elevate antioxidant responses and oxidative stress to a greater level than LLRE without BFR?

## Materials and methods

The protocol of the review was prospectively registered in PROSPERO, (CRD42020183204). Using the recommendations of Preferred Reporting Items for Systematic Reviews and Meta-analyzes (PRISMA-P) [19], a search was carried out in the electronic databases.

### Eligibility criteria

The research question was developed through the PICOS strategy: P—Healthy human subjects; I—resistance exercise with LLRE-BFR; C—LLRE or HLRE without BFR; O–oxidative stress biomarkers; S—randomized clinical trials. The studies were considered relevant based on the following criteria: studies performed with healthy subjects; low-load (defined as all loads between 20–50% of 1RM or maximal voluntary contraction [MVC] resistance exercises performed with BFR, compared with HLRE (defined as loads $\geq$ 70% 1RM /MVC) or LLRE without BFR (defined as load $\leq$ 50% 1RM / MVC) (1,3), evaluation of at least one oxidative stress biomarker and randomized clinical trials. Studies that were considered not relevant had subjects who received a substance before or after the intervention (i.e., supplementation) or the design of the study was not a clinical trial.

### Information sources

Seven databases were searched for eligible trials from inception until April 15, 2022: The Cochrane Central Register of Controlled Trials—CENTRAL, SPORTDiscus (Ebsco), EMBASE (Ebsco), PubMed, CINAHL (Ebsco) and Virtual Health Library- VHL, which includes Lilacs, Medline and SciELO, without restriction of year and language. Manual searches were conducted on the reference lists of the included records, as well as the documents that cited any of the included studies to identify potential eligible studies.

### Search strategy

The search in all databases was performed ('All field / All text") as shown and search strategies described in Supporting Information [S1 Search]. We conducted an additional search on

September 4, 2022 to identify potential studies published after April 15, 2022. This search strategy was used by two blinded researchers (JVF and MVF) independently. Disagreements were resolved through a third researcher (TDM) to reach a consensus.

## Selection process

The screening process was divided into three phases after implementing the search strategy. In the first phase, duplicate articles were eliminated using the Rayyan® software program [20]. In the second phase, a title and abstract search was performed by each reviewer and titles and abstracts that did not meet the eligibility criteria were excluded. In the last phase, all remaining articles considered relevant were assessed for eligibility based on full reading. In the included studies, the reference list was reviewed, and potentially relevant studies were assessed for eligibility to ensure all studies that met inclusion criteria were included.

## Data collection process

In this step, information about (1) the study authors and date; (2) population (i.e., number of subjects and their characteristics); (3) study design, (4) exercise protocol (number of series and repetitions, frequency of training, training length, exercised muscle, muscle action mode, and exercise load); (5) BFR cuff settings (pressure and width); and (5) biomarkers associated with oxidative stress measured in blood was extracted by each author (JVF and MVF) independently and disagreements were resolved by a third researcher (TDM). When different sampling times were used, the pre-exercise and the immediately post (until 5 minutes post-exercise), 24- and 48 hours after training session were considered. This was due to the lack of studies that had measured oxidative stress markers at consistent time points following the post-exercise.

If the values were presented in graph forms, Plot Digitizer 2.6.8 (Java, 2018) was used to perform data extraction, improving reliability between the data extracted by the reviewers [21].

## Risk of bias assessment

The quality assessment and potential bias of eligible studies was performed using the PEDro scale. According to the PEDro scale, clinical trials with a score <4 are considered 'poor', 4 or 5 'fair', 6, 7, or 8 'good,' and 9 or 10 'excellent'. For trials evaluating complex interventions (i.e., exercise), a PEDro score of 8/10 is considered optimal [22]. The study information was extracted independently by the researchers (JVF and MVF) for subsequent crosschecking of the data and discussion of possible discrepancies.

## Data synthesis and analysis

The meta-analysis was performed using RevMan software (version 5.4, Copenhagen, Denmark: The Nordic Cochrane Center, The Cochrane Collaboration, 2019) to summarize the effects of resistance training performed with BFR on pro-oxidant and antioxidant activity. For those studies evaluating multiple oxidative stress biomarkers, all were included. However, when possible, we performed a comparative meta-analysis for each biomarker. Data were pooled in meta-analyses and the effect size (ES) was described as standardized mean differences (SMD) when the data were presented in different outcome measures and as mean difference (MD) if the studies used the same outcome measure [23]. ES magnitudes of 0.2, 0.5, and 0.8 were classified as low, moderate and large, respectively [23,24]. Significance level was set at p<0.05 for all analyses.

For calculating ES, the changes (immediately-, 24- and 48 hours after exercise) from baseline (Mean post−Mean $_{pre}$) and standard deviations (SD) of pro-oxidants and antioxidants for all groups in each study were determined. All change values were estimated through calculations using the confidence interval calculator downloaded from the PEDro website [25]. If the SD information was not available, it was estimated from standard error, confidence interval or p-value according to the recommendations provided by Cochrane's Handbook for Systematic Reviews [23]. The analysis of pooled data was calculated using a random effect model because of the heterogeneity of the studies. Heterogeneity between studies was determined using the Cochrane's chi-squared test, and the percentage of variability in effect estimates that was attributable to heterogeneity rather than to chance was quantified using Higgins's inconsistency statistic ($I^2$), with thresholds set as $I^2 = 25\%$ (low), $I^2 = 50\%$ (moderate), and $I^2 = 75\%$ (high) [23].

In addition, we performed a subgroup analysis to explore the effects of different training models (HLRE and LLRE without BFR) compared to LLRE-BFR. When sufficient data was available, sensitivity analyzes were performed to assess the influence of different exercise protocols (i.e., failure versus fixed repetitions) on the effect estimated in the meta-analysis.

## Certainty of evidence

Two reviewers (JVF and TDM) assessed the quality of the current evidence using Grading of Recommendations, Assessment, Development and Evaluation (GRADE) methodology, regardless of whether there was enough information to perform the meta-analysis [26]. The classification of the quality of the evidence depends on five factors (risk of bias, inconsistency, indirectness, imprecision, publication bias) where for each factor not met, the quality of the evidence is reduced by one level (high to moderate, low or very low) [26]. The following factors were evaluated in each comparison to make a strength of evidence statement; Risk of bias was present if PEDro score < 6 in greater than 25% of the included trials; Inconsistency if $I^2 >$ 50%; Indirectness if > 50% of the subjects in the study were not related to the trial target's audience; Imprecision if < 400 subjects total in the comparison; publication bias was evaluated using a funnel plot when 10 or more studies were in the same comparison [26]. When comparisons of the activity of the oxidant/antioxidant biomarkers had only a single trial (<400 subjects), these studies were classified as inconsistent and imprecise, and were automatically downgraded to 'low quality evidence', which could be further downgraded to 'very-low quality evidence' if additional limitations were identified in relation to the risk of bias [27].

## Results

The search of seven databases provided a total of 256 citations. After removing the duplicates, the remaining articles were submitted to a title and abstract analysis and 18 articles were considered potentially eligible. Eight were subsequently excluded due to the following reasons: not using oxidative stress biomarkers [28,29], using aerobic exercise [30,31], the subjects had chronic kidney disease [32], overweight [33], received protein supplementation [34], or did not include a LLRE (≤50% of 1RM) or HLRE (≥70% of 1RM) group [35]. Three articles fulfilling the inclusion criteria were found after screening the reference list of the included studies [36–38]. Thus, 13 trials met all inclusion criteria (Fig 1) and were published between 2000–2021.

## Study selection and sample characteristics

The 13 studies in this review are summarized in Table 1 and included 158 subjects. Eight studies used a cross-over/inter-subject design [18,36–42] and five studies used a parallel design

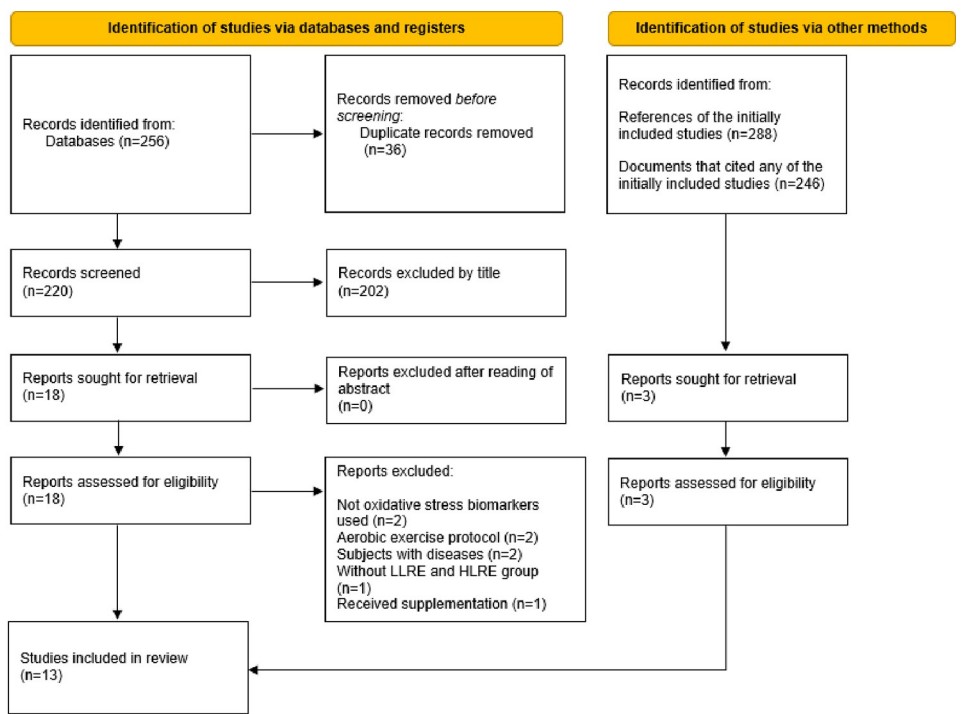

HLRE – High-load resistance exercise; LLRE – Low-load resistance exercise

**Fig 1. Flow diagram of trials in the review.** HLRE–High-load Resistance Exercise; LLRE–Low Load Resistance Exercise.

[43–47]. Included studies generally consisted of small sample sizes ranging from six [43] to twenty-eight subjects [47]. The studied populations were young male subjects between 18 and 35 years old. Five studies were composed of resistance trained subjects [18,36,37,40,41], four trials were conducted in physically / recreationally active men [38,39,44,47], one study classified its subjects as athletes [42], but provided no details about their training routine, while another three studies included sedentary/untrained subjects [43,45,46].

## Exercise protocol

Four of the thirteen studies performed both upper and lower limb resistance exercise [36,38,39,47] whereas five studies performed just upper [38,39,42,44,45], and four performed just lower limb exercises [18,41,44,45]. LLRE intensity varied between 20–50% 1RM while exercise in the HLRE varied between 70–80% 1RM. Of the 13 studies included in this review, six performed exercise to failure [36–39,44,45] but none of these trials reported equalizing training volume (total repetitions × load) between groups (LLRE-BFR versus HLRE or LLRE); one of them calculated training volume [45]. In this study, the LLRE-BFR group had ~2.3-fold greater training volume compared to HLRE. However, even without prescribing exercises until failure, Neto et al. [40] and Centner et al. [18] presented a higher training volume for the HLRE group of ~ 28% and 7%, respectively when compared to LLRE-BFR. Only two studies [41,43] attempted to equalize training volume between LLRE-BFR and LLRE during fixed repetition schemes, while for LLRE-BFR vs HLRE, two studies equalized the training volume [46,47]. All trials in this systematic review did not report any adverse events.

The most frequently used sampling time point for oxidative stress markers was immediately post-exercise (up to two minutes following completion) in twelve studies [18,36–40,42–47],

**Table 1. Characteristics of included trials.**

| Study | Design | Subjects, training status | Intervention | BFR cuff width / pressure | N | Exercise | Protocol Exercise |
|---|---|---|---|---|---|---|---|
| Boeno et al. [38] | Cross-over | Young physically active men 23.7 ± 3.4 years | LLRE-BFR: 30% 1RM with BFR HLRE: 80% 1RM | N/R / 20 mmHg above and below SBP for upper and lower limb, respectively | 11 | Bilateral elbow flexion and leg press | 4 sets to failure in each exercise with 1 minute of rest between sets |
| Centner et al. [18] | Cross-over | Healthy trained men; 24.8 ± 2.6 years | LLRE-BFR: 30% 1RM with BFR HLRE: 80% 1RM | 20 cm / 50% LOP | 15 | Front squats | LLRE-BFR: 1 set of 30 repetitions and 3 sets of 15 repetitions with 30 sec of rest; HLRE: 3 set of 10 repetitions |
| Garten et al. [37] | Cross-over | Resistance-trained men; 25 ± 3 years | LLRE-BFR: 30% 1RM with BFR HLRE: 70% 1RM | N/R / 20mmHg below SBP | 12 | Unilateral elbow flexion | 3 sets to failure with one-minute rest between sets |
| Goldfarb et al. [36] | Cross-over | Resistance-trained men; 21.3 ± 4.8 years | LLRE-BFR: 30% 1RM with BFR HLRE: 70% 1RM | 11 cm / 20mmHg below SBP | 7 | Unilateral elbow flexion and ankle plantarflexion | 3 sets to failure in each exercise with one-minute rest between sets |
| de Lima et al. [45] | Randomized clinical trial | Untrained men; 22.8 ± 2.1 years | LLRE-BFR: 20% 1RM with BFR HLRE: 75% 1RM | 6.5 cm / 50% LOP | 18 | Barbell arm curl | 6 sets until to failure with 90 seconds of rest between sets |
| Neto et al. [40] | Cross-over | Recreationally-trained men; 19 ± 0.8 years | LLRE-BFR: 20% 1RM with BFR HLRE: 80% 1RM | 6 cm / 130% SBP | 10 | Bilateral Bench press, front pulldown, triceps pressdown, biceps curl | LLRE-BFR: 1 set of 30 repetitions and 3 sets of 15 repetitions with 30 sec. rest between sets and 1 min interval between each exercise; HLRE: 3 sets of eight repetitions with 2-minute intervals between each series and a 1-minute intervals between each exercise |
| Nielsen et al. [44] | Randomized clinical trial | Recreationally active men 24 ± 3 years | LLRE-BFR: 20% 1RM with BFR HLRE: 70% 1RM | 13.5 cm / 100 mmHg | 20 | Unilateral knee extension | 4 sets to failure with 30 seconds rest for BFR and 90 seconds for HIT |
| Ramis et al. [39] | Cross-over | Physically active men; 23.72 ± 3.49 years | LLRE-BFR: 30% 1RM with BFR HLRE: 80% 1RM | 15 cm for upper, 17 for lower limb / 20 mmHg above and below SBP for upper and lower limb, respectively | 11 | Bilateral leg press and elbow flexion | 4 sets to failure in each exercise with 2 minutes of rest between sets |
| Ramis et al. [47] | Randomized clinical trial | Physically active men; 23.96 ± 2.67 years | LLRE-BFR: 30% 1RM with BFR HLRE: 80% 1RM | 14 cm for upper, 16 for lower limb / 20 mmHg below SBP for upper and 40 mmHg above the value used to occlude the upper limb | 28 | Unilateral elbow flexion and knee extension | LLRE-BFR: 4 sets of 21–23 repetitions for BFR HLRE: 4 sets of eight repetitions; both groups had 2 minutes of rest between sets |
| Ozaki et al. [46] | Randomized clinical trial | Young untrained males; 24 ± 2.64 years | LLRE-BFR: 30% 1RM with BFR HLRE: 75% 1RM | 3 cm / 100 to 160 mmHg | 14 | Bench press | LLRE-BFR: 4 sets (1 set of 30 repetitions followed by 3 sets of 15 repetitions with 30 seconds rest between sets HLRE: 3 sets of ten repetitions with 2–3 minutes between sets |
| **Summary of studies comparing LLRE-BFR versus LLRE** | | | | | | | |
| Boeno et al. [38] | Cross-over | Young physically active men; 23.7 ± 3.4 years | LLRE-BFR: 30% 1RM with BFR LLRE: 30% 1RM | N/R / 20 mmHg above and below SBP for upper and lower limb, respectively | 11 | Bilateral elbow flexion and leg press | 4 sets to failure in each exercise with an interval of 1 minute between sets |
| Centner et al. [18] | Cross-over | Healthy trained men; 24.8 ± 2.6 years | LLRE-BFR: 30% 1RM with BFR LLRE: 30% 1RM | 20 cm / 50% LOP | 15 | Front squats | 1 set of 30 repetitions followed by 3 sets of 15 repetitions with 30 seconds of rest between sets |

*(Continued)*

**Table 1.** (Continued)

| Study | Design | Subjects, training status | Intervention | BFR cuff width / pressure | N | Exercise | Protocol Exercise |
|---|---|---|---|---|---|---|---|
| Freitas et al. [41] | Cross-over | Trained men 20.58 ± 2.39 years | LLRE-BFR: 20% 1RM with BFR LLRE: 20% 1RM | 15 cm / 50%, 75% and 100% LOP | 12 | Unilateral knee extensions | 4 sets of 10 repetitions with 30 seconds of rest between sets |
| Garten et al. [37] | Cross-over | Resistance-trained men; 25 ± 3 years | LLRE-BFR: 30% 1RM with BFR LLRE: 30% 1RM | N/R / 20mmHg below SBP | 12 | Unilateral elbow flexion | 3 sets to failure with one-minute of rest between sets |
| Ramis et al. [43] | Randomized clinical trial | Sedentary men 24.57 ± 2.78 years | LLRE-BFR: 50% 1RM with BFR LLRE: 50% 1RM | N/R / 100 mmHg | 12 | Unilateral Elbow flexion | 3 sets of 15 repetitions with 30 seconds of rest between sets |
| Ramis et al. [39] | Cross-over | Physically active men 23.72 ± 3.49 years | LLRE-BFR: 30% 1RM with BFR LLRE: 30% 1RM | 15 cm for upper, 17 for lower limb / 20 mmHg above and below SBP for upper and lower limb, respectively | 11 | Bilateral leg press and elbow flexion | 4 sets to failure in each exercise with 2 minutes of rest between sets |
| Takarada et al. [42] | Cross-over | Young male athletes 20 ± 2 years | LLRE-BFR: 20% 1RM with BFR LLRE: 20% 1RM | 3 cm/ 214 ± 18.9 mmHg | 6 | Bilateral knee extension | 5 sets until failure with 30 seconds of rest between sets |

BFR: Blood flow restriction; 1RM: One repetition maximum, LLRE-BFR: Low-load resistance exercise with blood flow restriction; HLRE: High-load resistance exercise; LLRE: Low-load resistance exercise; SBP: Systolic blood pressure; LOP: Limb occlusion pressure; N/R: Not reported.

after 24 hours in four studies [39,41,44,45] and only two trials [39,41] sampled 48 hours post-exercise. When studies aimed to compare the effects of different degrees of limb occlusion pressure on oxidative stress markers [18,41,45], we only considered the 75% and 50% limb occlusion pressure groups to align with recent practice guidelines [6].

## Risk of bias

The risk of bias assessment is presented in Table 2. The PEDro score of the included trials ranged from 5 to 7 points (mean 5.53 points, SD = 0.77). Eight studies were classified with "fair quality" [18,36–40,42,43] while five were considered "good quality" [41,44–47]. All articles failed to blind the therapist and the subjects. However, the blinding of the therapist / subjects during the application of training with BFR is a methodological limitation due to the technique itself. Random allocation was not applied in three trials [18,36,37]. When there were dropouts, no trials performed intention-to-treat analysis, and only 15% of included studies [39,41] applied allocation concealment.

Among the 13 eligible trials, only two studies provided information on sample losses during the study methodology [39,40]. In addition, no study reported existence of a prospectively registered protocol; only three reported making sample size calculations to adequately power analyses [40,41,46].

## LLRE-BFR produces lower lipid peroxidation and protein oxidation compared to HLRE, but not compared to LLRE

The evaluation of oxidation products in biological molecules was used to measure oxidative damage generated to lipids [39–42,45] and proteins [36,37,39,40,43,45]. Other biomarkers

**Table 2. PEDro scores of included studies.**

| Study | 2 | 3 | 4 | 5 | 6 | 7 | 8 | 9 | 10 | 11 | Total |
|---|---|---|---|---|---|---|---|---|---|---|---|
| Boeno et al. [38] | Y | N | Y | N | N | N | Y | N | Y | Y | 5 |
| Centner et al. [18] | N | N | Y | N | N | N | Y | Y | Y | Y | 5 |
| Freitas et al. [41] | Y | Y | Y | N | N | N | Y | Y | Y | Y | 7 |
| Garten et al. [37] | N | N | Y | N | N | N | Y | Y | Y | Y | 5 |
| Goldfarb et al. [36] | N | N | Y | N | N | N | Y | Y | Y | Y | 5 |
| de Lima et al. [45] | Y | N | Y | N | N | N | Y | Y | Y | Y | 6 |
| Neto et al. [40] | Y | N | Y | N | N | N | N | Y | Y | Y | 5 |
| Nielsen et al. [44] | Y | N | Y | N | N | N | Y | Y | Y | Y | 6 |
| Ozaki et al. [46] | Y | N | Y | N | N | N | Y | Y | Y | Y | 6 |
| Ramis et al. [39] | Y | Y | Y | N | N | N | N | N | Y | Y | 5 |
| Ramis et al. [43] | Y | N | Y | N | N | N | Y | Y | Y | Y | 5 |
| Ramis et al. [47] | Y | N | Y | N | N | Y | Y | Y | Y | Y | 7 |
| Takarada et al. [42] | N | N | Y | N | N | N | Y | Y | Y | Y | 5 |

Y: Yes; N: No.

analyzed were sulfhydryl / thiols [39,43,45] and nitric oxide [38,46,47]. The measurement of nitric oxide in all studies was evaluated by concentration of nitrite and nitrate in the blood, as an index of endothelial nitric oxide synthase activity. Centner et al. [18] measured the total ROS generation with LLRE-BFR while Garten et al. [37] evaluated the activity of xanthine oxidase.

Of the included studies, three trials (n = 59) [39,40,45] compared the post-exercise effect of lipid peroxidation between LLRE-BFR and HLRE. Our results, based on low certainty of evidence (downgraded due imprecision and risk of bias), demonstrates that LLRE-BFR produces significantly lower post-exercise lipid peroxidation products than HLRE with a large effect size (SMD = -0.95 CI 95%: -1.49 to -0.40, $I^2$ = 0%, p = 0.0007), see Fig 2. Only one trial (n = 11) analyzed changes in lipid damage markers 24 hours and 48 hours after exercise [39]. This study reported that LLRE-BFR produces significantly lower lipid peroxidation products (p = 0.02) at 24 hours post-exercise, but at the 48-hour sampling point, no statistically significant difference was reported (p > 0.05), see Table 3. All these findings are based on very-low certainty of evidence (downgraded due imprecision, inconsistency, and risk of bias). On the other hand, based on three studies comparing the effects of LLRE-BFR with LLRE, no significant differences were observed in lipid damage markers at any time point evaluated in this review with low certainty of evidence (downgraded due to risk of bias and imprecision). For more details, see Figs 2 and 3.

Five trials (n = 87) [36,37,39,40,45] assessed immediate post-exercise damage to proteins in LLRE-BFR versus HLRE. Our analysis provides very-low certainty of evidence (downgraded due imprecision, inconsistency, and risk of bias) that LLRE-BFR generates lower protein damage products when compared with HLRE, with a large effect size (SMD = -1.39 CI 95%: -2.11 to -0.68, $I^2$ = 51%, p = 0.0001), see Fig 4. In addition, three studies (n = 55) [37,39,43] demonstrate LLRE-BFR promotes no difference compared with LLRE post-exercise (SMD = -0.43 CI 95%: -0.97 to 0.11, $I^2$ = 0%, p = 0.12). These effects were based on low certainty of evidence (downgraded due imprecision and risk of bias). At other time points, we did not observe any differences between LLRE-BFR versus HLRE or LLRE (see Table 3).

Finally, we did not observe any differences between protocols in concentrations of sulfhydryl / thiols [39,43,45]. Furthermore, three pooled studies (n = 62) [38,46,47] provides low certainty (downgraded due imprecision and inconsistency) that LLRE-BFR promotes similar

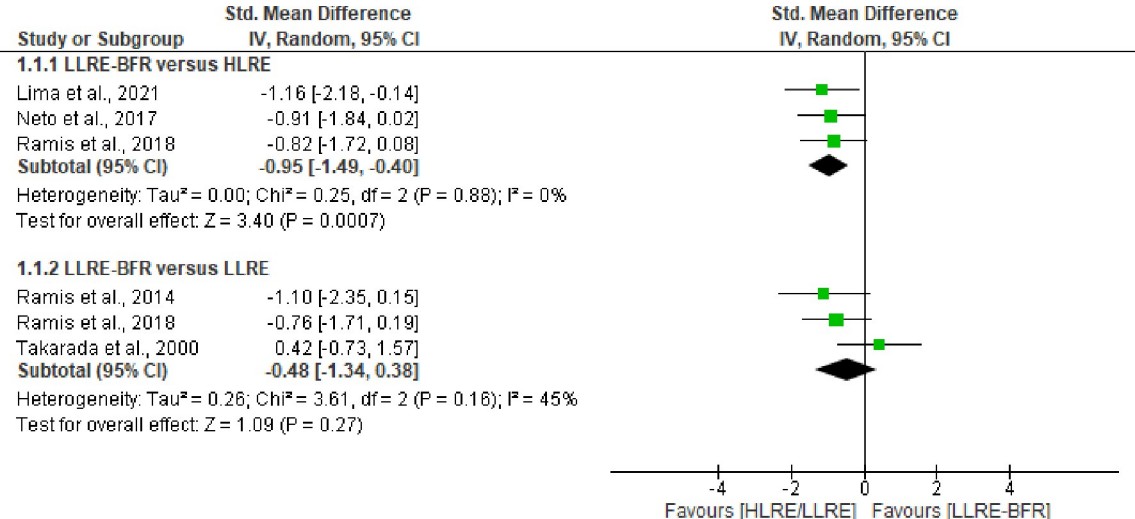

**Fig 2. Forest plot about effects of LLRE-BFR versus HLRE and LLRE on biomarkers of damage to lipids post-exercise.**
LLRE-BFR: Low load resistance exercise with blood flow restriction; HLRE: High-load resistance exercise, LLRE: Low load resistance exercise; SMD: Standardized mean difference; SD: Standard deviation; IV: Inverse variance; CI: Confidence interval.

levels of NO as HLRE (SMD = 0.31 CI 95%: -0.20 to 0.80, $I^2$ = 4%, p = 0.23), see Fig 5. In the other biomarkers and time points, none of the studies showed significant differences between LLRE-BFR versus HLRE and LLRE-BFR versus LLRE (see Table 3).

**Effects of LLRE-BFR compared with HLRE and LLRE in endogenous antioxidants.** Studies measured antioxidant enzymes from superoxide dismutase [38,39] and catalase [38,39,43], as well as non-enzymatic antioxidants total glutathione [36] and uric acid [39,40,43]. In addition, three trials verified the total antioxidant capacity [41,44,45]. During data extraction, we observed that superoxide dismutase activity presented by Ramis et al. [39] was included in another study [38]. Therefore, we considered only the results from Ramis et al. [39], avoiding mistaken conclusions from duplication of data and conforming to recommendations of Cochrane's Handbook for Systematic Reviews [23].

Superoxide dismutase activity was attenuated after LLRE-BFR compared to HLRE immediately post-exercise, and 24 hours and 48 hours post-exercise [39]. When compared with LLRE, LLRE-BFR presents similar responses in all post-exercise sampling times [39]. Catalase activity was measured in three studies [38,39,43]. The pooled results of two trials (n = 43) [38,39] provided low certainty of evidence (downgraded due imprecision and risk of bias) that there was not differences between LLRE-BFR and HLRE post-exercise (MD = -0.25 U catalase mg CI 95%: -1.81 to 1.30, $I^2$ = 0%, p = 0.75). Three studies [38,39,43] (n = 53) showed similar effect between LLRE-BFR and LLRE (MD = 0.15 U catalase/mg CI 95%: -2.87 to 3.16, $I^2$ = 62%, p = 0.92), see Fig 6. The certainty of evidence was rated as very low (downgraded due imprecision, inconsistency, and risk of bias). Only one trial (n = 11) [38] compared catalase activity changes after LLRE-BFR versus HLRE or LLRE at 24 hours and 48 hours post exercise and demonstrates no differences between them (Table 3).

Four studies (n = 62) [37,41,44,45] measured total antioxidant capacity activity, see Table 3. Total antioxidant capacity was measured through trolox equivalent antioxidant capacity assay [41,44], the oxygen radical absorbance capacity assay [37], and the plasma iron reduction ability [45]. Three trials compared LLRE-BFR versus HLRE immediately post-exercise [37,44,45] and one study 24 hours post-exercise [44]. LLRE-BFR versus LLRE was evaluated immediately [37], 24 hours [41] and 48 hours [41] post-exercise.

**Table 3. Description of study results in LLRE-BFR versus HLRE and LLRE on antioxidant and oxidative biomarkers based on changes (post–pre-exercise).**

| Study | Time-points | Antioxidant Biomarker | Main Results | Oxidative Biomarker | Main Results |
|---|---|---|---|---|---|
| Boeno et al. [38] | Post-exercise | SOD# CAT | HLRE ↑ the levels of SOD compared with ↓ LLRE-BFR, but in LLRE versus LLRE-BFR there was not difference. There was not difference between groups in CAT activity | NOx | LLRE-BFR showed significant ↑ levels of NOx compared with HLRE, but not LLRE |
| Centner et al. [18] | Post-exercise | - | - | Total ROS | No difference between conditions |
| Freitas et al. [41] | 24 hours and 48 hours post exercise | TAC | No difference between LLRE-BFR and LLRE all time-points | Lipid Peroxide | No difference between LLRE-BFR and LLRE all time-points |
| Garten et al. [37] | Post-exercise | ORAC GSH status | GSH status was significant ↑ in HLRE compared with LLRE-BFR, but not in LLRE-BFR versus LLRE. No difference between groups was observed in ORAC. | CP Xanthine Oxidase [XO] | CP ↑ increase in HLRE compared with LLRE-BFR, but not has difference between LLRE-BFR and HLRE or LLRE in XO. |
| Goldfarb et al. [36] | Post-exercise | GSH status | GSH status was ↑ in HLRE compared with baseline, but not significant compared to BFR | CP | CP was ↑ in BFR and HLRE, but not significant differences |
| de Lima et al. [45] | Post-exercise | Iron Reduction Ability GSH ratio | GSH ratio was significantly ↓ in HLRE compared with LLRE-BFR, Iron reduction ability was ↓ in HLRE group | TBARS Thiols CP | TBARS and CP ↑ in HLRE, but not in LLRE-BFR. There was no significant effect of on thiol groups between groups. |
| Neto et al. [40] | Post-exercise | Uric acid | No significant difference in LLRE-BFR and HLRE | TBARS CP | No significant differences between groups |
| Nielsen et al. [44] | Post-exercise and 24 hours | TAC Total GSH | No difference in TAC for LLRE-BFR and HLRE, GSH ↑ in HLRE | - | No significant difference between groups |
| Ozaki et al. [46] | Post-exercise | - | - | NOx | No significant differences between groups |
| Ramis et al. [39] | Post -exercise, 24 hours and 48 hours | SOD CAT | SOD and CAT activity was ↑ in HLRE compared with LLRE-BFR post 24 hours, but not in other moments. No differences in SOD between LLRE-BFR and LLRE in all time points. CAT ↑ in LLRE-BFR post- exercise and 48 hours compared with LLRE. | TBARS; carbonylated protein CP; Sulfhydryl | TBARS ↑ post 24 hours in HLRE compared with LLRE-BFR, but no between group differences between LLRE and LLRE-BFR |
| Ramis et al. [43] | Post-exercise | CAT Uric acid | There was no difference between LLRE-BFR and LLRE | TBARS; carbonylated protein CP; Sulfhydryl | No significant between group differences |
| Ramis et al. [47] | Post-exercise | - | - | NOx | No significant between group differences |
| Takarada et al. [42] | Post-exercise and 24 hours | - | - | Lipid peroxidase | No significant between group differences |

SOD: Superoxide dismutase; TAC: Total Antioxidant Capacity; GSH: Glutathione; LLRE-BFR: Low-load with resistance exercise with blood flow restriction; HLRE: High-load resistance exercise; LLRE: Low load resistance exercise; BFR: Blood flow restriction; TBARS: Thiobarbituric acid-reactive substance; CP: Carbonylated proteins; XO: Xanthine Oxidase; ORAC: Oxygen radical absorbance capacity; NOx: Nitric oxide; CAT: Catalase.

No differences between LLRE-BFR and LLRE were observed in any moments. Although three studies measured total antioxidant capacity immediately post-exercise in HLRE and LLRE-BFR, it was not possible to perform a meta-analysis. One study [42] did not provide data necessary to be included in a meta-analysis, even after we contacted the corresponding

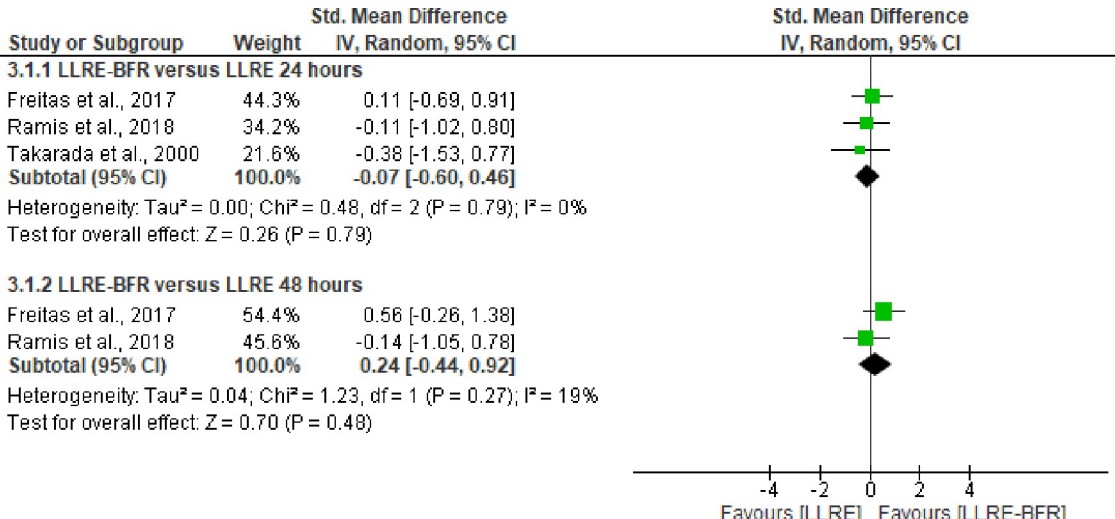

**Fig 3. Forest plot about effects of LLRE-BFR versus LLRE on biomarkers of damage to lipids 24 hours and 48 hours post-exercise.** LLRE-BFR: Low load resistance exercise with blood flow restriction; HLRE: High-load resistance exercise, LLRE: Low load resistance exercise; SMD: Standardized mean difference; SD: Standard deviation; IV: Inverse variance; CI: Confidence interval.

author via e-mail. Due to the high level of heterogeneity ($I^2 > 75\%$), we did not combine the results of the two remaining studies, but results are presented in Table 3 and provide low certainty of evidence (downgraded due imprecision and inconsistency).

The trial conducted by Nielsen et al. [44] reported an increase in total glutathione (reduced glutathione + oxidized glutathione) concentration immediately post-exercise in the HLRE group, but unchanged in LLRE-BFR group immediately- and 24 hours post-exercise. However, Lima et al. [45] demonstrated no difference between groups regarding reduced glutathione and oxidized glutathione levels post-exercise. Finally, Neto et al. [40] and Ramis et al. [43]

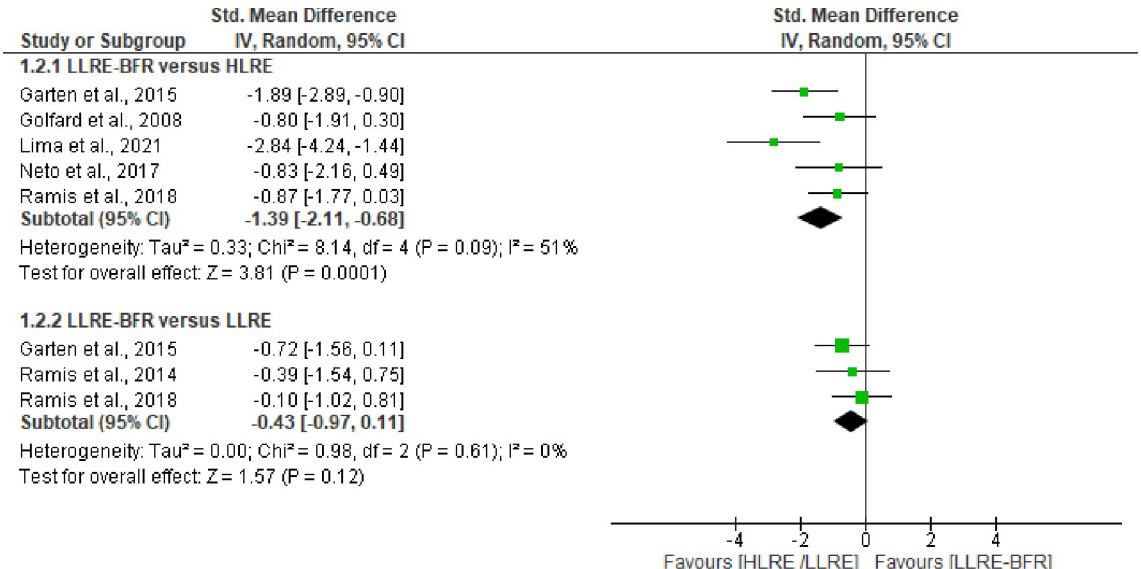

**Fig 4. Forest plot about effects of LLRE-BFR versus HLRE and LLRE on biomarkers of damage to proteins post-exercise.** LLRE-BFR: Low load resistance exercise with blood flow restriction; HLRE: High-load resistance exercise, LLRE: Low load resistance exercise; SMD: Standardized mean difference; SD: Standard deviation; IV: Inverse variance; CI: Confidence interval.

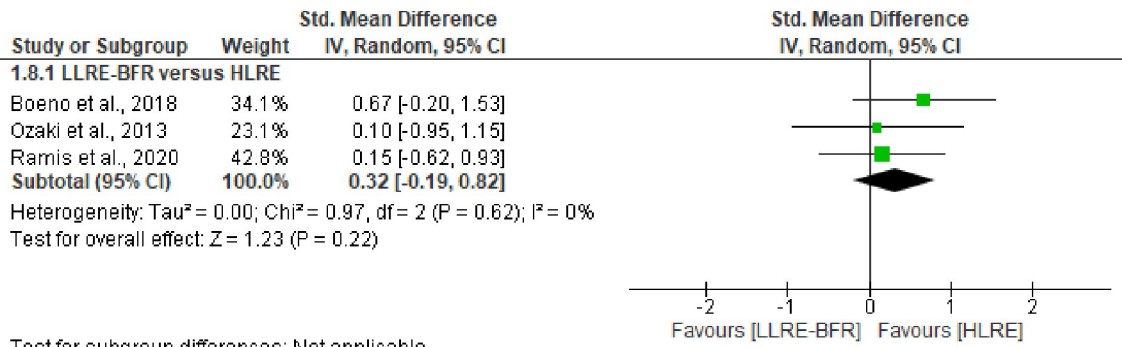

**Fig 5. Forest plot about effects of LLRE-BFR versus HLRE on biomarkers of production of NO post-exercise.** LLRE-BFR: Low load resistance exercise with blood flow restriction; HLRE: High-load resistance exercise, LLRE: Low load resistance exercise; SMD: Standardized mean difference; SD: Standard deviation; IV: Inverse variance; CI: Confidence interval.

demonstrated that uric acid levels did not show any significant differences between LLRE-BFR, LLRE and HLRE.

**HLRE causes changes in the glutathione redox ratio biomarker, but LLRE-BFR does not.** Three trials (n = 37) [36,37,44] evaluated redox balance through glutathione redox ratio immediate post-exercise. The pooled results demonstrates that LLRE-BFR reduces the gluta-thione redox ratio with a large effect size (SMD = -1.12 CI 95%: -1.70 to -0.55, $I^2$ = 0%, p = 0.0001) compared with HLRE, see Fig 7. When comparing LLRE-BFR with LLRE, there are no significant differences (p = 0.66). The certainty of evidence in this comparison was clas-sified as low (downgraded due imprecision and risk of bias).

**Certainty of evidence assessment.** The overall certainty of evidence for each biomarker in each comparison is presented in the S1 Table. In summary, the level of certainty of evidence ranged from low to very low due to imprecision, heterogeneity, or risk of bias. As none of the meta-analyses included more than 10 studies, Egger's test could not be used to assess publica-tion bias. Therefore, we assessed publication bias by evaluating the search strategy and use of industry funding; the results appear to indicate that none of the meta-analyses were affected by publication bias in this manner.

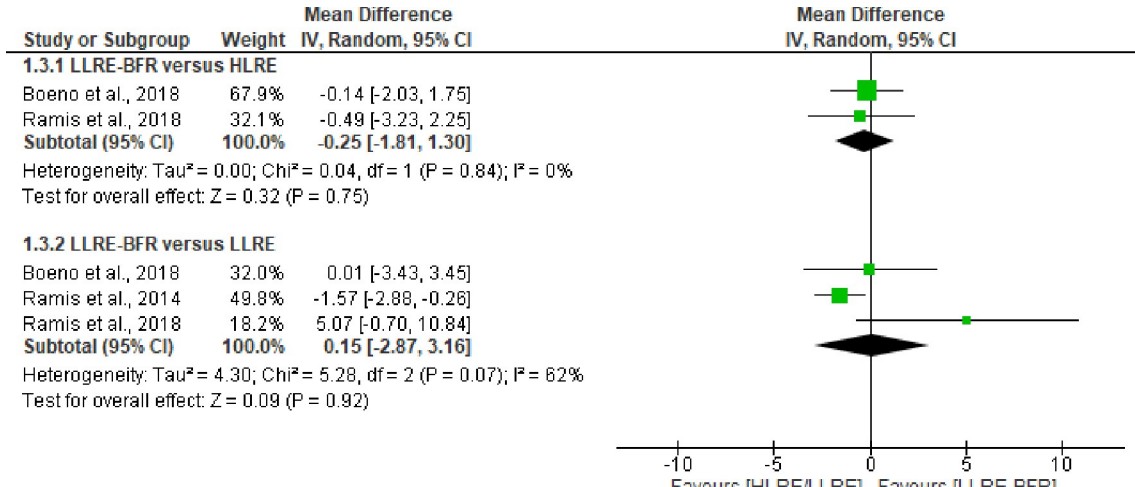

**Fig 6. Forest plot about effects of LLRE-BFR versus HLRE and LLRE on catalase activity post-exercise.** LLRE-BFR: Low load resistance exercise with blood flow restriction; HLRE: High-load resistance exercise, LLRE: Low load resistance exercise; MD: Mean difference; SD: Standard deviation; IV: Inverse variance; CI: Confidence interval.

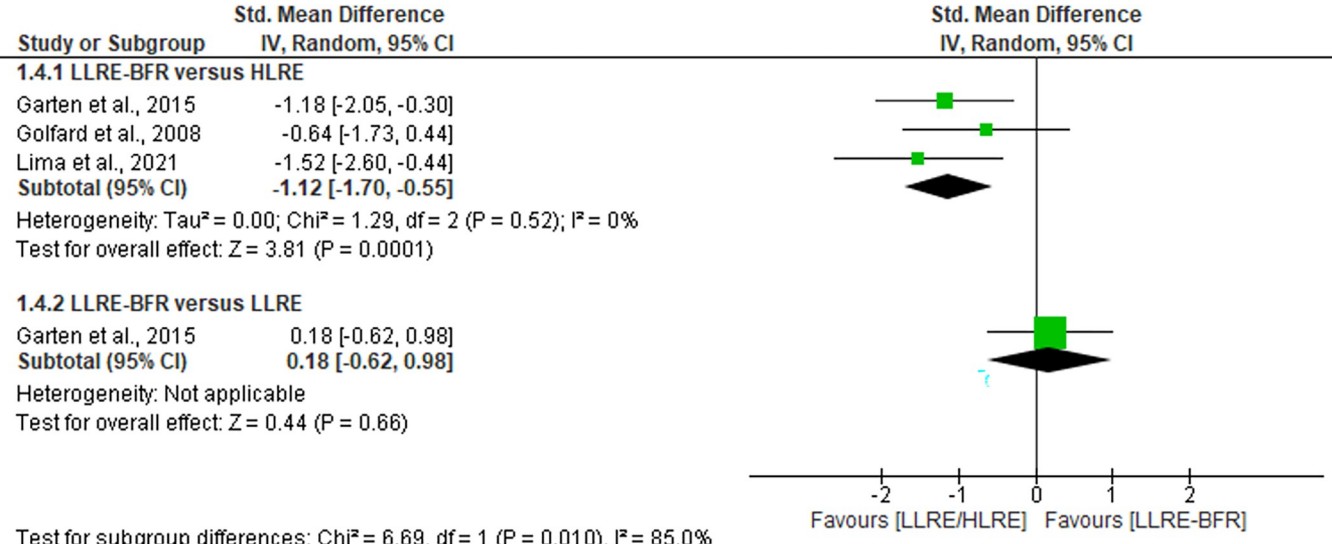

**Fig 7. Forest plot about effects of LLRE-BFR versus HLRE on glutathione redox ratio post-exercise.** LLRE-BFR: Low load resistance exercise with blood flow restriction; HLRE: High-load resistance exercise, LLRE: Low load resistance exercise; SMD: Standardized mean difference; SD: Standard deviation; IV: Inverse variance; CI: Confidence interval.

## Discussion

This systematic review with meta-analysis aimed to examine available scientific evidence about the effects of an acute session of LLRE-BFR on changes in oxidative stress biomarkers compared with traditional HLRE and LLRE. To the best of our knowledge, this is the first systematic review with meta-analysis on the topic. Our data suggests that an acute LLRE-BFR session promotes lower levels of lipid peroxidation and protein damage compared to HLRE and similar levels when compared to LLRE. On the other hand, one trial reported that an acute LLRE-BFR session promotes lower superoxide dismutase activity, but in remaining trials, similar responses in other biomarkers such as catalase and uric acid were recorded compared to HLRE. In addition, HLRE appears to promote greater redox imbalance in favor of oxidants than LLRE-BFR. For LLRE-BFR compared to LLRE, regardless of repetition scheme utilized (failure or non-failure), no differences were observed. Therefore, our data does not support the idea that reactive oxygen species (ROS) formation induced by LLRE-BFR are central in the upregulated signal transduction pathways responsible for muscle remodeling and for the adaptive responses observed during LLRE-BFR, at least with respect to the data collected in the studies included within this review.

Because LLRE-BFR can promote comparable adaptations to traditional HLRE despite a lower load [48], it has been hypothesized that increased ROS production may mediate this response [6]. ROS responses are thought to be due to the metabolic stress caused by hypoxia/ischemia in the muscle that may be potentiated by the venous restriction. Given that, we questioned whether a LLRE-BFR session would promote similar oxidative stress-induced damage than HLRE or LLRE.

Recently Groennebaek et al. [49] demonstrated that LLRE-BFR produced similar mitochondrial adaptations (i.e., elevated mitochondrial protein synthesis and respiratory function) following training theorized to be related to increased ROS production despite LLRE-BFR using a markedly lower load compared to HLRE. However, despite some trials exercising subjects to volitional failure, there was not a significant increase in markers of oxidative muscle damage after LLRE-BFR [37,45]. In contrast, Rodrigues et al. [17] demonstrated that forearm

ischemic exercise generated increases in oxidation products (lipids concentrations) immediately post-exercise. However, after one-minute post-exercise, concentrations returned to their baseline value indicating that these elevations were transient. Petrick et al. [50] corroborates these findings, showing both maximal and submaximal mitochondrial ROS emission rates in human skeletal muscle tissue were acutely decreased 2 hours following LLRE-BFR, but not after HLRE. With respect to our data, our pooled analyses indicates that an acute session of LLRE-BFR promotes lower levels of lipid peroxidation and protein damage markers compared to HLRE. Accordingly, previous investigations have shown that resistance exercise with high intensities (load) can increase ROS production, regardless of training volume [51,52]. Therefore, ROS production in resistance training in non-failure protocols is primarily load dependent.

Conversely, the levels of oxidative markers found in this review after LLRE-BFR was similar to LLRE. However, LLRE-BFR induces earlier fatigue compared to LLRE [53]. Thus, it is possible that when subjects exercised to failure in LLRE [37,39,42], the metabolic responses were similar between conditions [54] and may explain the similar antioxidant and oxidative stress responses. Another study by Kolind et al. [55] supports this hypothesis and sheds light on possible similarities between mechanisms underlying LLRE with- and without BFR. Their research concluded that when both LLRE and LLRE-BFR are performed to failure, and normalized to time to task failure, muscle oxygenation, muscular excitability and blood pooling were similar despite LLRE-BFR performing ~43% less work. It is possible that in the lower volume studies (40 to 45 repetitions) conducted by Ramis et al. [43] and Freitas et al. [41], the presence or absence of BFR was insufficient to change redox status, producing no significant changes in concentration of biomarkers associated with oxidative damage and antioxidant defense. Thus, it appears that performing repetitions closer to failure may be a critical factor for inducing redox imbalance, that is, higher levels of ROS–although this is speculative and likely requires intra-exercise measurements to confirm or refute.

A question that remains is, if ROS production is not the pivotal mechanism to LLRE-BFR-induced muscle adaptation, how may it cause comparable adaptations to traditional HLRE? Centner et al. [18] suggests that LLRE-BFR may positively influence muscle remodeling and adaptations according to the hormesis principle [56]. The hormesis hypothesis proposes that beneficial muscle adaptations occur at an optimal ROS level but begin to become deleterious with additional accumulations, leading to various types of cellular damage declines in exercise-induced adaptation [57,58].

Other mechanisms associated with ROS production that contribute to increased strength and hypertrophy through LLRE-BFR are attributed increased rates of muscle protein synthesis, activating pathways similar to those activated after high-load strength training (i.e., signaling mammalian target of rapamycin complex-1 [mTORC1] and mitogen activated protein kinase) [59]. In this sense, studies conducted by Fry et al. [60] and Fujita et al. [61] demonstrated that exercise with BFR can increase protein synthesis and mTORC1 and mitogen activated protein kinase signaling pathways in elderly and young individuals, respectively. A recent study [59] compared the acute and chronic molecular and muscular hypertrophic and strength responses following a 7-week knee flexor/knee extensor resistance training program performed with LLRE-BFR and HLRE. Their research indicated that muscular hypertrophy and strength of the knee extensors/knee flexors increased similarly while the acute and molecular responses (i.e., mTORC1, ERK ½, JNK Phos/Total) following exercise were largely identical despite significantly different volumes (~43%) and metabolic stress. Therefore, enhanced anabolic molecular signaling may be an important cellular mechanism that may partly explain the hypertrophy induced by LLRE-BFR. However, some evidence suggests that ROS can activate or inhibit mTORC1. For example, there is evidence [62] demonstrating that low dose, short-term ROS

exposure stimulates mTORC1 while high concentration, long-term exposures of ROS inhibits mTORC1 activity.

Indeed, this leads us to suggest that time of ischemia/BFR and the magnitude of loads used during LLRE-BFR likely determine ROS production during resistance training. The stimulus of training with BFR usually lasts from 10 to 20 minutes [6,7]. It is likely that the magnitude of the loads used (20–50% 1RM) together with the shorter time in local tissue ischemia, produce a stimulus that is unable to significantly increase the ROS concentrations associated with the aforementioned negative outcomes (i.e., myonecrosis). Previous studies support that the additional metabolic stress induced by LLRE-BFR does not have an additive effect when the resistance protocol is performed with high-load [16,63,64]. One of the possible mechanisms proposed in the literature to explain these results is that the reperfusion following the ischemic stimulus increases nitric oxide (NO) bioavailability and modulates mitochondrial oxygen consumption, leading to a reduction in ROS production and less oxidative damage to the cell [65]. Thus, it is possible that cuff deflation can generate increased vascular shear stress by positively regulating NO-induced vasodilation to promote additional venous blood flow which, in turn, may contribute to elimination of ROS [66]. Indeed, the effects of LLRE-BFR on NO metabolism are not yet fully understood, however the pooled data in our analysis [37,45,46] suggest that LLRE-BFR has a similar effect to HLRE in NO production.

Although able to generate lower levels of lipid and protein damage when compared to HLRE, one clinical trial demonstrated LLRE-BFR produced lower superoxide dismutase (SOD) activity when compared to HLRE, while we observed similar effects regarding endogenous antioxidants. This may seem expected given the greater increase in lipid, protein damage markers, and cellular redox imbalances promoted by HLRE. Greater SOD activity probably occurred to catalyze the dismutation of superoxide radicals ($O_2^{-\bullet}$) and the formation of hydrogen peroxide ($H_2O_2$). As a consequence of SOD activity, $H_2O_2$ is generated and dismutated by catalase or glutathione peroxidase (GPx) [67]. The action of these enzymes varies in relation to the concentration of $H_2O_2$, as GPx is more effective in low concentrations [58]. Considering that no change in catalase enzyme activity was observed after exercise comparing LLRE-BFR and HLRE, it is possible that GPx dismutated and formed $H_2O_2$. However, none of the studies included in this review evaluated $GP_X$ activity. Furthermore, included studies that measured antioxidant capacity showed heterogeneous results. It's possible that the divergence of results may be impacted by the training status of the subjects and the impact of ROS overload on total antioxidant capacity. This partly explains our results, since one study with untrained subjects showed a greater post-exercise redox imbalance in favor of HLRE but not in LLRE-BFR [42].

It is well established that trained individuals are less susceptible to oxidative damage than untrained individuals due to a greater antioxidant defense capacity [68]. Therefore, untrained subjects are less effective at neutralizing ROS generated during exercise and may indicate that antioxidant enzymes are regulated by the redox status and not by the exercise itself, as suggested by Bessa et al. [14]. However, the current literature does not allow us to draw a firm conclusion on whether training status can influence LLRE-BFR-mediated redox responses. Investigating the effects of LLRE-BFR training on oxidative stress markers in both untrained and trained populations could further elucidate the relevancy of ROS and subsequent oxidative stress to the adaptation processes following LLRE-BFR.

Furthermore, there are no studies comparing the effects of the LLRE-BFR protocol to muscle failure versus a predefined repetition protocol on oxidative stress markers, but we observed that protocols conducted to failure or non-failure with LLRE-BFR were less likely to induce oxidative stress than HLRE. In contrast, we observed a tendency in HLRE to induce oxidative stress regardless of training status [42,44]. Finally, studies elucidating the effects of a long-term LLRE-BFR protocol on the modulation of redox system responses are needed and will help to understand the magnitude of the impact of LLRE-BFR on ROS generation.

The findings of this review can help coaches and professionals working in rehabilitation. Our findings suggest that LLRE-BFR can be a potentially useful strategy in exercise prescription as LLRE-BFR seems to attenuate ROS-induced cell damage and injury when compared to HLRE in healthy individuals. LLRE-BFR is mainly relevant in periods of intensive training and in the rehabilitation of musculoskeletal injuries (i.e., post-surgery conditions, in those suffering from arthritis, or in elderly frail individuals), where the chemical and enzymatic antioxidant defense system may be insufficient to combat the rise of free radicals induced by exercise [69,70]. This has important implications in clinical situations, since disturbances in redox homeostasis in favor of oxidants are associated with delayed muscle recovery, greater delayed onset muscle soreness, fatigue, and reduction in strength, affecting muscle performance [14,69].

In this way, the exercise-induced oxidative stress could likewise be relevant for exercise prescription. Although a HLRE session induces greater oxidative damage, it seems unlikely that this training modality amplifies oxidative stress enough to be harmful to human health given the many physiologic benefits observed with chronic training programs. In light of this, LLRE-BFR is considered a generally safe and effective resistance exercise approach that may aid in faster muscle recovery and similar benefits to HLRE given adoption of appropriate application parameters (i.e., load, pressure, and time under BFR) and high levels of effort during training [6]. However, compared to HLRE, LLRE with- or without BFR reduces ROS levels and could be a feasible strategy in practice if minimizing oxidative stress is desired.

## Limitations and strength points

This systematic review and meta-analysis are not without limitations. It is important to note that the data included within this review comes from studies with methodological shortcomings and different forms of application of the BFR stimulus, highlighting the large heterogeneity within the current body of literature on LLRE-BFR. Most articles included within this review were considered to have fair quality with a mean PEDro score of ~5.53. Secondly, we included only up to three time points (post-, 24 hours, 48 hours) of oxidative marker responses. Conclusions on the time-course of oxidative stress outside of this period are scarce. Thirdly, we did not include studies that used aerobic exercise in this review, so the effects of BFR associated with aerobic exercise on oxidative stress biomarkers remain unclear. Future studies should ensure more rigorous research methodologies (i.e., follow the CONSORT checklist, description of size sample, etc.), including reporting and implementing all BFR-related prescriptions (i.e., justify the pressures of cuff, reporting cuff widths and whether the occlusive stimulus was applied throughout the entire exercise or deflated between rest periods) that are considered standard of practice.

The strengths of this review include a detailed search strategy using multiple databases and manual searches through manuscript reference lists without any publication or language restrictions. Second, this systematic review was pre-registered in the International Prospective Register of Systematic Reviews (PROSPERO) and all stages of this review followed PRISMA-P recommendations. Third, the quality of the evidence was carefully evaluated according to the GRADE approach. Last, we evaluated the effects of low-load exercise associated with LLRE-BFR compared to HLRE and LLRE on each biomarker separately, an approach that has not been performed systematically in any prior manuscripts.

## Conclusion

In summary, based on low or very low certainty evidence, a LLRE-BFR session generates less oxidative damage and short-term (48 hours) redox imbalances than a HLRE session in healthy

subjects. No significant differences in oxidative damage concentration and antioxidant response after exercise with LLRE-BFR compared to LLRE were observed. Furthermore, the results of our systematic review and meta-analysis suggest that exercise-mediated ROS production is likely not a central mechanism for BFR-induced physiological adaptations such as muscle hypertrophy and muscle strength gain.

## Supporting information

**S1 Checklist. PRISMA checklist.**
(DOCX)

**S1 Table. Table certainty of evidence (GRADE system).** GRADE: Grades of Recommendation, Assessment, Development and Evaluation; SMD: Standardized mean difference; MD: Mean difference; BFR: Blood flow restriction HLRE: High-load resistance exercise LLRE: Low-load resistance exercise GSSG: Oxidized glutathione GSH: Reduced glutathione*More than 25% of participants from studies with low methodological quality (Physiotherapy Evidence Database score < 6 points).# Whether more 50% of participants were not similar to those about whom conclusions are drawn ($I^2 > 50\%$)† 75% of participants or less from studies with findings in the same direction.‡ Fewer than 400 participants for each outcome. n/a: Not applicable; was not performed due to the insufficient number of studies (<10 studies).
(DOCX)

**S1 Search. Search strategy.**
(DOCX)

## Author Contributions

**Conceptualization:** João Vitor Ferlito, Nicholas Rolnick, Thiago De Marchi, Mirian Salvador.

**Data curation:** João Vitor Ferlito, Marcos Vinicius Ferlito.

**Formal analysis:** João Vitor Ferlito, Marcos Vinicius Ferlito.

**Methodology:** João Vitor Ferlito, Marcos Vinicius Ferlito, Thiago De Marchi, Rafael Deminice.

**Software:** João Vitor Ferlito, Thiago De Marchi.

**Supervision:** Thiago De Marchi, Rafael Deminice, Mirian Salvador.

**Validation:** Nicholas Rolnick.

**Visualization:** João Vitor Ferlito, Nicholas Rolnick, Thiago De Marchi, Rafael Deminice, Mirian Salvador.

**Writing – original draft:** João Vitor Ferlito, Nicholas Rolnick, Thiago De Marchi, Rafael Deminice, Mirian Salvador.

**Writing – review & editing:** João Vitor Ferlito, Nicholas Rolnick, Rafael Deminice.

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
