## [Decision Letter · Decision Letter 0]

12 Dec 2022

PONE-D-22-31577Acute Effect of Low Load Resistance Exercise with Blood Flow Restriction on Oxidative Stress Biomarkers: A Systematic Review and Meta-AnalysisPLOS ONE

Dear Dr. Ferlito,

Thank you for submitting your manuscript to PLOS ONE. After careful consideration, we feel that it has merit but does not fully meet PLOS ONE’s publication criteria as it currently stands. Therefore, we invite you to submit a revised version of the manuscript that addresses the points raised during the review process.

ACADEMIC EDITOR:Dear authors,

The work is verry interesting and it has potential to be published in Plos One. However, instructions for authors of Plos One were not followed. Moreover there are some methodological issues that need to be amended. It is highly recommeded to check 

The PRISMA 2020 statement: an updated guideline for reporting systematic reviews, BMJ 2021;372:n71. http://dx.doi.org/10.1136/bmj.n71

For instance, the author will be able to observe that the order of the methods section is not correct. Also, there are some sections that need to be reformulated or amended. In addition, please address comments of both reviewers. 

Please carefully review all these issues. 

Thank you  

We look forward to receiving your revised manuscript.

Kind regards,

Rafael Franco Soares Oliveira

Academic Editor

PLOS ONE

“The author(s) received no financial support for the research, authorship, and/or publication of this article.”

“NR is the founder of THE BFR PROS, a BFR education company that provides BFR training workshops to fitness and rehabilitation professionals across the world using a variety of BFR devices. NR has no financial relationships with any cuff manufacturers/distributors.  The remaining authors declare that they have no conflict of interests.”

5. Please include your tables as part of your main manuscript and remove the individual files. Please note that supplementary tables (should remain/ be uploaded) as separate "supporting information" files

Additional Editor Comments:

Dear authors,

The work is verry interesting and it has potential to be published in Plos One. However, instructions for authors of Plos One were not followed. Moreover there are some methodological issues that need to be amended. It is highly recommeded to check

The PRISMA 2020 statement: an updated guideline for reporting systematic reviews, BMJ 2021;372:n71. http://dx.doi.org/10.1136/bmj.n71

For instance, the author will be able to observe that the order of the methods section is not correct. Also, there are some sections that need to be reformulated or amended. In addition, please address comments of both reviewers.

Please carefully review all these issues.

Thank you

Reviewers' comments:

Reviewer's Responses to Questions

**Comments to the Author**

1. Is the manuscript technically sound, and do the data support the conclusions?

Reviewer #1: Yes

Reviewer #2: Partly

2. Has the statistical analysis been performed appropriately and rigorously? 

Reviewer #1: Yes

Reviewer #2: Yes

3. Have the authors made all data underlying the findings in their manuscript fully available?

Reviewer #1: Yes

Reviewer #2: Yes

4. Is the manuscript presented in an intelligible fashion and written in standard English?

Reviewer #1: Yes

Reviewer #2: Yes

5. Review Comments to the Author

Reviewer #1: The manuscript was duly revised and adjustments were made by the authors, and minor adjustments are necessary for publication, this is our technical opinion. The theoretical framework is outdated and there are still inconsistencies regarding their formatting.

Reviewer #2: Dear Authors,

I would like to express my gratitude regarding the opportunity to review this manuscript.

It is an interesting study, congratulations. At this stage the manuscript requires considerable improvements. Below suggestions with line or page indication.

5-30 - Please review the instructions for authors and change the text format accordingly.

34-54 - Please revise the abstract text, namely the abbreviations format and sections (methods, results, and others), considering the instructions for authors.

36 - “high load” / 62 - “high-load” - Please carefully revise all manuscript standardizing these details.

58 - Please review the instructions for authors, subtopics normally do not present numbers in this journal.

70 and 100 - The reference format is different. Please correct in all cases throughout the manuscript.

Please review line 133.

149-150 - I do not think “XXX” is adequate. It is more important to understand the authors involved in the search academic background and experience.

160, 168, 169, 177, 182 - Please consider changing “XXX”.

174-175 - Please standardize the text format “-“ or “/”.

202 - Please include city and country.

224 - Please provide a reference.

243 - Please include end point.

269 - “et al.” - end point missing. Same in lines 390, 429, 432, 433, 517 and other lines (please revise all manuscript).

298 - In the same line, numbers in different format. Please standardize throughout the manuscript.

303 - Please reformulate the text.

336, 342, 349 - Please include end points. This should be considered in all cases throughout the manuscript.

336, 342, 349 - Please consider text between figures and table.

387 - Please correct “[40,41,]”.

511 – “Kolind [54]” - incorrect citation format. Please correct.

556 - It seems more than one space after end point, please revise.

586-604 - This and other paragraphs are too long and do not favor reading. Please consider splitting.

633-635 - “secondly” and “thirdly” suggested.

663 - After conclusions section and before references, please consider the journal template, namely including Author Contribution.

666 – Please carefully revise all references format and change according to the instructions for author’s format.

Page 36 – Figure 1 - Some incorrections seem to be present in figure. The “identification” box, the position of “=” is not standardized and aside of “included” box it seems to appear a word track change. Please carefully revise the figure content.

38 - All the other figures should be corrected, namely including “et al.” and improving the figures quality. The figures footers (legends) should also be carefully revised.

Please carefully revise the English throughout the manuscript.

6. PLOS authors have the option to publish the peer review history of their article (what does this mean?). If published, this will include your full peer review and any attached files.

Reviewer #1: **Yes: **Felipe J. Aidar

Reviewer #2: No

---

## [Author Response · Author response to Decision Letter 0]

12 Jan 2023

It is copy/pasted here, but will be better viewed in the document as it is color coded.

Thank you to both reviewers for their comments on our manuscript. We have taken the time to go through and address all comments and in the process, reshaped the manuscript significantly based on your feedback. We wish to thank both of you for the time spent and the thoughtful comments provided to help make a more cohesive document. Our responses are in red for each of the specific comments. For your convenience, when appropriate/relevant, we have copied/pasted the specific sections of the manuscript for reference and these are presented in green. We hope that these changes make the article suitable for publication and thank you again!

Reviewer #1: The manuscript was duly revised and adjustments were made by the authors, and minor adjustments are necessary for publication, this is our technical opinion. The theoretical framework is outdated and there are still inconsistencies regarding their formatting.

General comments:

Conclusion

Are presented satisfactorily. However, I should have the practical applications of what I found.

In the new revised document, the practical applications are discussed in lines 612-631 whereas our conclusion is specifically related to the observations gleaned from our analysis. In short, we comment on the fact that LLRE-BFR may be important for those with injuries or pathologies where an imbalance in redox status is likely occurring and how LLRE-BFR does not appear to induce additional imbalances in redox status, compared to HLRE and as such, could be a suitable alternative for those populations. This is particularly true given longitudinal studies have shown LLRE-BFR induces similar muscle hypertrophy and strength gains as HLRE.

The information is copy-pasted here for your reference:

“The findings of this review can help coaches and professionals working in rehabilitation. Our findings suggest that LLRE-BFR can be a potentially useful strategy in exercise prescription, since LLRE-BFR seems to attenuate ROS-induced cell damage and injury when compared to HLRE in healthy individuals. This is mainly relevant in periods of intensive training and in the rehabilitation of musculoskeletal injuries (i.e., post-surgery conditions, in those suffering from arthritis, or in elderly frail individuals), where the chemical and enzymatic antioxidant defense system may be insufficient to combat the rise of free radicals induced by exercise (70,71). This has important implications in clinical situations, since disturbances in redox homeostasis in favor of oxidants are associated with delay muscle recovery, greater delayed onset muscle soreness, fatigue, and reduction in strength, affecting the muscle performance (14,70). 

In this way, the exercise-induced oxidative stress could likewise be relevant for exercise prescription. Although a HLRE session induces greater oxidative damage, it seems unlikely that this training modality amplifies oxidative stress enough to be harmful to human health given the many physiologic benefits observed with chronic training programs. In light of this, LLRE-BFR is considered a generally safe and effective resistance exercise approach that may aid in faster muscle recovery since certain thresholds are not surpassed (i.e., load, pressure, and time under BFR) reducing the propensity to accumulate ROS while still stimulating positive musculoskeletal outcomes comparable to HLRE.”

References

68 references were presented, their numbering is incorrect, the formatting needs to be revised, and 30 are outdated. please review.

Thank you for the comments. We have gone into the document and completely redid our references, reformatted according to the journal’s specifications and reviewed all of our citations to determine redundancy. We kept most of them because they are relevant and provide evidence support for the statements made about reactive oxygen species and do not appear to be outdated. We did include a couple of newer references that we did not cite originally when we went back to revise the manuscript. We hope that the revisions are suitable and if you believe that newer references are still required, we will make the appropriate adjustments if possible. There are now 71 references.

Overview

The manuscript presented addresses a relevant research topic.

It would be advisable to do a general review. 

Thank you for the comments. We have taken them into consideration and have substantially revised the manuscript for language, formatting, and content/analysis. We hope you view the new version of this manuscript favorably as we feel it is considerably improved based on your feedback.

Reviewer #2: Dear Authors,

I would like to express my gratitude regarding the opportunity to review this manuscript.

It is an interesting study, congratulations. At this stage the manuscript requires considerable improvements. Below suggestions with line or page indication.

5-30 - Please review the instructions for authors and change the text format accordingly

The manuscript was carefully revised in accordance with journal standards. We hope that the changes are in accordance with your request.

34-54 - Please revise the abstract text, namely the abbreviations format and sections (methods, results, and others), considering the instructions for authors.

We have made the appropriate changes as per journal specifications. We apologize for not meeting formatting requirements on our initial submission. Thank you.

For your convenience, the abstract is presented here:

Abstract

Background

The purpose of this review was to analyze the acute effects of low-load resistance exercise with blood flow restriction (LLE-BFR) on oxidative stress markers in healthy individuals in comparison with LLE or high-load resistance exercise (HLRE) without BFR. 

Materials and Methods

A systematic review was performed in accordance with the PRISMA (Preferred Reporting Items for Systematic Reviews and Meta-Analyses) guidelines. These searches were performed in CENTRAL, SPORTDiscus, EMBASE, PubMed, CINAHL and Virtual Health Library- VHL, which includes Lilacs, Medline and SciELO. The risk of bias and quality of evidence were assessed through the PEDro scale and GRADE system, respectively. 

Results

Thirteen randomized clinical trials were included in this review (total n=158 subjects). Results showed lower post-exercise damage to lipids (SMD= -0.95 CI 95% -1.49 to -0. 40, I2=0%, p=0.0007), proteins (SMD= -1.39 CI 95% -2.11 to -0.68, I2=51%, p=0.0001) and redox imbalance (SMD= -0.96 CI 95% -1.65 to -0.28, I2=0%, p=0.006) in favor of LLRE-BFR compared to HLRE. HLRE presents higher post-exercise superoxide dismutase activity but in the other biomarkers and time points, no significant differences between conditions were observed. For LLRE-BFR and LLRE, we found no difference between the comparisons performed at any time point. 

Conclusions

Based on the available evidence from randomized trials, providing very low or low certainty of evidence, this review demonstrates that LLRE-BFR promotes less oxidative stress when compared to HLRE but no difference in levels of oxidative damage biomarkers and endogenous antioxidants between LLRE.

Register number: PROSPERO number: CRD42020183204

36 - “high load” / 62 - “high-load” - Please carefully revise all manuscript standardizing these details.

Thank you for noticing the differences. We have standardized “high-load” as the phrase throughout the manuscript. 

58 - Please review the instructions for authors, subtopics normally do not present numbers in this journal.

We have made the appropriate adjustments to adhere to the formatting of the journal.

70 and 100 - The reference format is different. Please correct in all cases throughout the manuscript.

We have standardized the format to Vancouver citations throughout the manuscript as per the journal’s specifications 

Please review line 133.

Ln 133 is the search strategy figure. We are unsure of what you would like us to do. Please clarify if this comment is still valid. In the new manuscript, it occurs on Ln 260:

Figure 1. Flow diagram of trials in the review. HLRE – High-load Resistance Exercise; LLRE – Low Load Resistance Exercise.

149-150 - I do not think “XXX” is adequate. It is more important to understand the authors involved in the search academic background and experience.

160, 168, 169, 177, 182 - Please consider changing “XXX”.

We have made changes to the manuscript as requested and any new text is highlighted in red. We also added the researchers initials for the reviewer. They are presented in text here:

Ln 159-61: This search strategy was used by two blinded researchers (JVF and MVF) independently. Disagreements were resolved through a third researcher (TDM) to reach a consensus.

Ln 179-81: (5) biomarkers associated with oxidative stress measured in blood was extracted by each author (JVF and MVF) independently and disagreements were resolved by a third researcher (TDM).

Ln 194: The study information was extracted independently by the researchers (JVF and MVF)

Ln 230: Two reviewers (JVF and TDM) assessed the quality of the current evidence using

174-175 - Please standardize the text format “-“ or “/”.

We have made the changes to standardize it to remove ambiguity. In the revised version of the manuscript, it is on Ln 192: considered 'poor', 4 or 5 'fair', 6, 7, or 8 'good,' and 9 or 10 'excellent'. For trials evaluating

202 - Please include city and country. 

We have added the country as the city (Copenhagen) was already inside the citation. We hope this meets your revision requirements and it is on Ln 200 in the revised manuscript: Copenhagen, Denmark: The Nordic Cochrane Center, The Cochrane Collaboration,

224 - Please provide a reference.

We have added a citation to support the Higgin’s inconsistency measure. It is reference 23 in text and presented on Ln 221: thresholds set as I2 = 25% (low), I2 = 50% (moderate), and I2 = 75% (high) (23).

243 - Please include end point.

We have made the adjustment on Ln 261 in the revised manuscript: LLRE – Low Load Resistance Exercise. 

269 - “et al.” - end point missing. Same in lines 390, 429, 432, 433, 517 and other lines (please revise all manuscript).

We have added the end point as requested and made similar adjustments throughout the manuscript and are highlighted in red. We hope this meets your revision request.

298 - In the same line, numbers in different format. Please standardize throughout the manuscript.

We have made the adjustments and ensured the same formatting and can be found on line 858 in the new manuscript.

303 - Please reformulate the text.

We have made the changes according to the specifications of the journal and it is now within the body of the manuscript starting on line 318.

336, 342, 349 - Please include end points. This should be considered in all cases throughout the manuscript.

We have made the appropriate changes requested throughout the manuscript.

336, 342, 349 - Please consider text between figures and table.

We moved Figure 2 to begin at Line 332. This provides some text between all figures/tables. Thank you. 

387 - Please correct “[40,41,]”.

We have made the adjustments in the revision. It is shown here on Ln 400 in the revised manuscript: catalase (38,39,43), as well as non-enzymatic antioxidants total glutathione (36) and uric

511 – “Kolind [54]” - incorrect citation format. Please correct.

We have made the adjustments in the revision. It is now cited correctly on Ln 522: antioxidant and oxidative stress responses. Another study by Kolind et al. (56) supports

556 - It seems more than one space after end point, please revise.

We actually removed this from the manuscript as it was not needed given some of the new edits.

586-604 - This and other paragraphs are too long and do not favor reading. Please consider splitting.

We split the paragraph in half to facilitate a better reading experience. Starting on Ln 596-613 it reads:

It is well established that trained individuals are less susceptible to oxidative damage than untrained individuals due to a greater antioxidant defense capacity (69). Therefore, untrained subjects are less effective at neutralizing ROS generated during exercise and may indicate that antioxidant enzymes are regulated by the redox status and not by the exercise itself, as suggested by Bessa et al. (14). However, the current literature does not allow us to draw a firm conclusion on whether training status can influence LLRE-BFR-mediated redox responses. Investigating the effects of LLRE-BFR training on oxidative stress markers in both untrained and trained populations could further elucidate the relevancy of ROS and subsequent oxidative stress to the adaptation processes following LLRE-BFR. 

Furthermore, there are no studies comparing the effects of the LLRE-BFR protocol to muscle failure versus a predefined repetition protocol on oxidative stress markers, but we observed that protocols conducted to failure or non-failure with LLRE-BFR were less likely to induce oxidative stress than HLRE. In contrast, we observed a tendency in HLRE to induce oxidative stress regardless of training status (42,44). Finally, studies elucidating the effects of a long-term LLRE-BFR protocol on the modulation of redox system responses are needed and will help to understand the magnitude of LLRE-BFR in the generation of ROS.

633-635 - “secondly” and “thirdly” suggested.

We made the changes as requested and they’re presented on Ln 641-643: PEDro score of ~5.53. Secondly, we included only up to three time points (post-, 24-, 48 hours) of oxidative marker responses. Conclusions on the time-course of oxidative stress outside of this period are scarce. Thirdly, we did not include studies that used aerobic

663 - After conclusions section and before references, please consider the journal template, namely including Author Contribution.

We have added this on Ln 672-686.

666 – Please carefully revise all references format and change according to the instructions for author’s format.

All changes have been made to meet journal reference specifications.

Page 36 – Figure 1 - Some incorrections seem to be present in figure. The “identification” box, the position of “=” is not standardized and aside of “included” box it seems to appear a word track change. Please carefully revise the figure content.

We made all the changes requested in Figure 1.

38 - All the other figures should be corrected, namely including “et al.” and improving the figures quality. The figures footers (legends) should also be carefully revised.

We made all the changes requested in Figure 1 as well as in all other figures (adding et al. as well as making sure all figures are congruent. We hope the changes are sufficient. We also removed all captions in the figure section as this is not within journal guidelines. 

Please carefully revise the English throughout the manuscript.

The manuscript was carefully and extensively revised, I hope that the changes are in accordance with your request.

---

## [Decision Letter · Decision Letter 1]

3 Feb 2023

PONE-D-22-31577R1Acute effect of low-load resistance exercise with blood flow restriction on oxidative stress biomarkers: A systematic review and meta-analysisPLOS ONE

Dear Dr. Ferlito,

Thank you for submitting your manuscript to PLOS ONE. After careful consideration, we feel that it has merit but does not fully meet PLOS ONE’s publication criteria as it currently stands. Therefore, we invite you to submit a revised version of the manuscript that addresses the points raised during the review process.

ACADEMIC EDITOR: Dear authors,

There are still some minor issues to fix and a major one which is related the risk of bias assessment missing plots.

Please address all the last comments made by reviewers.

Thank you in advance

Please submit your revised manuscript by Mar 20 2023 11:59PM If you will need more time than this to complete your revisions, please reply to this message or contact the journal office at plosone@plos.org. Please include the following items when submitting your revised manuscript:A rebuttal letter that responds to each point raised by the academic editor and reviewer(s). You should upload this letter as a separate file labeled 'Response to Reviewers'.A marked-up copy of your manuscript that highlights changes made to the original version. You should upload this as a separate file labeled 'Revised Manuscript with Track Changes'.An unmarked version of your revised paper without tracked changes. You should upload this as a separate file labeled 'Manuscript'.If applicable, we recommend that you deposit your laboratory protocols in protocols.io to enhance the reproducibility of your results. Protocols.io assigns your protocol its own identifier (DOI) so that it can be cited independently in the future. For instructions see: https://journals.plos.org/plosone/s/submission-guidelines#loc-laboratory-protocols. Additionally, PLOS ONE offers an option for publishing peer-reviewed Lab Protocol articles, which describe protocols hosted on protocols.io. Read more information on sharing protocols at https://plos.org/protocols?utm_medium=editorial-email&utm_source=authorletters&utm_campaign=protocols.

We look forward to receiving your revised manuscript.

Kind regards,

Rafael Franco Soares Oliveira

Academic Editor

PLOS ONE

Journal Requirements:

Additional Editor Comments :

Dear authors,

There are still some minor issues to fix and a major one which is related the risk of bias assessment missing plots.

Please address all the last comments made by reviewers.

Thank you in advance

Reviewers' comments:

Reviewer's Responses to Questions

**Comments to the Author**

1. If the authors have adequately addressed your comments raised in a previous round of review and you feel that this manuscript is now acceptable for publication, you may indicate that here to bypass the “Comments to the Author” section, enter your conflict of interest statement in the “Confidential to Editor” section, and submit your "Accept" recommendation.

Reviewer #2: All comments have been addressed

Reviewer #3: All comments have been addressed

Reviewer #4: All comments have been addressed

2. Is the manuscript technically sound, and do the data support the conclusions?

Reviewer #2: Partly

Reviewer #3: Yes

Reviewer #4: Yes

3. Has the statistical analysis been performed appropriately and rigorously? 

Reviewer #2: Yes

Reviewer #3: Yes

Reviewer #4: Yes

4. Have the authors made all data underlying the findings in their manuscript fully available?

Reviewer #2: Yes

Reviewer #3: Yes

Reviewer #4: Yes

5. Is the manuscript presented in an intelligible fashion and written in standard English?

Reviewer #2: No

Reviewer #3: Yes

Reviewer #4: Yes

6. Review Comments to the Author

Reviewer #2: Dear Authors,

Thank you for considering my suggestions and incorporating them into the manuscript, which globally improved, congratulations. The suggestion for minor revision is close.

Below suggestions related to this last version, with line indication.

11, 13, 17 – Please revise the text format throughout the manuscript. One example in these 3 lines, the text starts with different formats.

72 and 73 – Please correct the numbers and text format.

119 – “and” seems appropriated before “2)”. Please consider.

136 – It is suggested MVC in full in the first appearance in the manuscript, but afterwards, in lines 138 and 139, only the abbreviation.

173 – It seems a line spacing is missing, please revise.

256 – “(39,41,43).”, although, citation 36-42 only in line 267. Please carefully revise because this compromises all the reference and citation numbers.

262, 263 – It looks like more than one line spacing, please revise.

274-275 – I believe word track changes emerge close to line number, please revise.

299 – Please revise the table content. Some text alignment in central, other not. Standardization according to the journal instructions for authors is suggested. Also, the text paragraph size is not standardized, please carefully revise. Please also revise all tables format (lines and other details).

301 – All abbreviations should be in full in table footnote.

318 - Some text alignment in central, other not. Standardization according to the journal instructions for authors is suggested in table 2.

319 – Please revise text format.

355 – Table 1 footnote presents end point, table 2 not, please standardize considering the journal instructions for authors

431 – “24 hours” suggested instead of “24-“.

479/480 – “heavy load resistance exercise (HLRE)” 460 – “High-load resistance exercise”. Please revise all manuscript considering the correction of these details.

660 – Please insert line spacing.

Please carefully revise all references format. For example, ref 1 is incorrect. “internet” and link unnecessary, also volume and number in wrong format, “30(15)” in this particular example.

Please carefully revise the English throughout the manuscript.

Please carefully revise all manuscript considering the journal instructions for authors. For example, in the citations, square brackets seem to be required by the journal.

Reviewer #3: Dear authors,

All the points raised by he previous reviewers were satisfactory answered by your team. This manuscript follows the higher standards in conducting systematic reviews and addresses a topic of interest to the readers and to scientific community.

I have no further comments to add.

Best regards,

Reviewer #4: In this study, Joao and colleagues systematically evaluated the evidence on exercise-induced oxidative stress in resistance exercise with and without BFR and demonstrated that HLRE promotes oxidative stress more than LLRE-BFR. Also, they didn’t find any significant difference in levels of oxidative damage biomarkers and endogenous antioxidants between LLRE BFR and LLRE. Despite the results being quite interesting, the manuscript requires some revisions.

1- The reviewed oxidative stress markers are not included in the abstract

2- The keywords have not been selected properly

3- The introduction is too long, Also, it should be mentioned what kind of oxidative stress markers are induced by exercise and what are the mechanisms of inducing them.

4- Instead of TBARS please mention MDA

5- In addition to number of papers please add the number of subjects.

6- Publication bias evaluation is important and to investigate the presence of publication bias in the meta-analysis, the funnel plot, Begg’s rank correlation and Egger’s weighted regression tests can be used. Please add funnel plot analysis.

7. PLOS authors have the option to publish the peer review history of their article (what does this mean?). If published, this will include your full peer review and any attached files.

Reviewer #2: No

Reviewer #3: No

Reviewer #4: No

---

## [Author Response · Author response to Decision Letter 1]

8 Feb 2023

Although the use of the Cochrane Risk of Bias tool provides a plot summarizing the studies' risk of bias judgments, we did not use it in this review. According to the previously registered review protocol on PROSPERO, we used the PEDRO scale. Furthermore, this rating scale is recommended for use in meta-analysis to assess risk of bias. If you wish us to use something else, please let us know, but we wanted to follow our protocol and felt the PEDRO scale adequately assessed risk of bias in this meta-analysis (Doi & Barendregt, 2013) and has been recommended as a substitute in lieu Cochrane Risk of Bias as long as there is consistency (Moseley et al., 2019). 

Of note, other meta-analyses have successfully been published in PlosONE using PEDRO scale for bias assessment without Cochrane risk of bias assessment: https://journals.plos.org/plosone/article?id=10.1371/journal.pone.0264557

https://journals.plos.org/plosone/article?id=10.1371/journal.pone.0100503

Doi, S. A. R., & Barendregt, J. J. (2013). Not PEDro’s bias: summary quality scores can be used in meta-analysis [Review of Not PEDro’s bias: summary quality scores can be used in meta-analysis]. Journal of Clinical Epidemiology, 66(8), 940–941. Elsevier BV.

Moseley, A. M., Rahman, P., Wells, G. A., Zadro, J. R., Sherrington, C., Toupin-April, K., & Brosseau, L. (2019). Agreement between the Cochrane risk of bias tool and Physiotherapy Evidence Database (PEDro) scale: A meta-epidemiological study of randomized controlled trials of physical therapy interventions. PloS One, 14(9), e0222770.

Reviewer #2

11, 13, 17 – Please revise the text format throughout the manuscript. One example in these 3 lines, the text starts with different formats.

Our response: Thank you, this has been amended.

72 and 73 – Please correct the numbers and text format.

Our response: Thank you, this has been amended.

119 – “and” seems appropriated before “2)”. Please consider.

Our response: Thank you, this has been amended.

136 – It is suggested MVC in full in the first appearance in the manuscript, but afterwards, in lines 138 and 139, only the abbreviation.

Our response: Thank you, this has been amended.

173 – It seems a line spacing is missing, please revise.

Our response: Thank you, this has been amended.

256 – “(39,41,43).”, although, citation 36-42 only in line 267. Please carefully revise because this compromises all the reference and citation numbers. 

Our response: Thank you, this has been amended to reflect accurate citation numbers.

262, 263 – It looks like more than one line spacing, please revise.

Our response: Thank you, this has been amended.

274-275 – I believe word track changes emerge close to line number, please revise.

Our response: Thank you, this has been amended.

299 – Please revise the table content. Some text alignment in central, other not. Standardization according to the journal instructions for authors is suggested. Also, the text paragraph size is not standardized, please carefully revise. Please also revise all tables format (lines and other details).

Our response: Thank you, this has been amended.

301 – All abbreviations should be in full in table footnote.

Our response: Thank you, this has been amended (line 305). 

318 - Some text alignment in central, other not. Standardization according to the journal instructions for authors is suggested in table 2.

Our response: Thank you, this has been amended.

319 – Please revise text format.

Our response: Thank you, this has been amended.

355 – Table 1 footnote presents end point, table 2 not, please standardize considering the journal instructions for authors.

Our response: Thank you, this has been amended.

431 – “24 hours” suggested instead of “24-“.

Our response: Thank you, this has been amended.

479/480 – “heavy load resistance exercise (HLRE)” 460 – “High-load resistance exercise”. Please revise all manuscript considering the correction of these details.

Our response: Thank you, this has been amended.

660 – Please insert line spacing.

Our response: Thank you, this has been amended.

Please carefully revise all references format. For example, ref 1 is incorrect. “internet” and link unnecessary, also volume and number in wrong format, “30(15)” in this particular example.

Our response: We have removed all “[internet]” mentions in the references list. We formatted our references in accordance with “Vancouver” format as recommended by the journal’s guidelines so we are unsure 

Please carefully revise the English throughout the manuscript.

Our response: We have a primary English speaker as a lead author and have gone through the manuscript in another round of revisions. Any changes are made in red throughout the manuscript with track changes. We feel the manuscript articulates our points effectively, but if you feel there is specific sections that are not clear after these revisions, can you please point them out so we can further scrutinize to improve our communication of these topics? Thank you.

Reviewer #3: 

Dear authors,

All the points raised by he previous reviewers were satisfactory answered by your team. This manuscript follows the higher standards in conducting systematic reviews and addresses a topic of interest to the readers and to scientific community.

I have no further comments to add.

Best regards,

Our response: Our sincere thanks for your review and the positive feedback on the article.

Reviewer #4: 

1- The reviewed oxidative stress markers are not included in the abstract

Our response: In this review, we observed that more than 10 different biomarkers were used to assess oxidative stress. Therefore, we chose to summarize the main findings of the review in the abstract. Thus, we followed the guidelines of PLOS ONE regarding the writing of the abstract.

The Abstract should: " (a) Describe the main objective(s) of the study; (b) Explain how the study was done, including any model organisms used, without methodological detail; (c) Summarize the most important results and their significance; (d) Not exceed 300 words"

2- The keywords have not been selected properly

Our response: Thank you, this has been amended.

“Keywords: reactive oxygen species, free radicals, ischemic training, exercise, resistance training”

3- The introduction is too long, Also, it should be mentioned what kind of oxidative stress markers are induced by exercise and what are the mechanisms of inducing them.

Our response: Thank you for your comment, we added a sentence to help readers better understand this topic.

“Furthermore, after resistance exercise, physiological responses such as the infiltration of phagocytes (ie, neutrophils and macrophages) at the site of injury are necessary for neuromuscular adaptation. This exercise-induced inflammatory response also contributes to increased free radical production, contributing to oxidative damage to biomolecules.”

However, in our introduction, we approach the relationship between oxidative stress and exercise in a summarized way, since the literature presents a diversity of tests and markers that can be used to measure the levels of oxidative stress. Furthermore, exercise-induced ROS production comes from different sources, as described in the introduction. In our view, addressing each biomarker and the mechanism involved would make the introduction extremely long, not following the journal's recommendations.

Thus, in the introduction, we chose to maintain the journal's recommendations: "Provide a background that contextualizes the manuscript and allows readers outside the area to understand the purpose and meaning of the study, as well as a brief review of the literature."

4- Instead of TBARS please mention MDA

Our response: Although both the TBARS and MDA assays are both used to estimate the lipid damage caused by the increase in reactive species induced by muscle activity during exercise, it is already well evidenced in the literature that the TBARS assay can interact with a variety of oxidized lipids, both saturated and unsaturated aldehydes, sucrose and urea to form various chromogens, thus not specifically measuring just MDA. For this reason and based on the current literature, we chose not to make changes to the table referring to oxidative stress markers.

Forman et al, 2015: “Thus, one cannot directly equate the measurement of TBARS with MDA or lipid peroxidation when measured in a complex biological system.”

Forman HJ, Augusto O, Brigelius-Flohe R, et al. Even free radicals should follow some rules: a guide to free radical research terminology and methodology. Free Radic Biol Med. 2015;78:233-235. doi:10.1016/j.freeradbiomed.2014.10.504

Moselhy HF, Reid RG, Yousef S, Boyle SP. A specific, accurate, and sensitive measure of total plasma malondialdehyde by HPLC. J Lipid Res. 2013;54(3):852-858. doi:10.1194/jlr.D032698

5- In addition to number of papers please add the number of subjects.

Our response: Thank you, this has been amended throughout the manuscript. 

6- Publication bias evaluation is important and to investigate the presence of publication bias in the meta-analysis, the funnel plot, Begg’s rank correlation and Egger’s weighted regression tests can be used. Please add funnel plot analysis.

Our response: We partial agree with the reviewer, however as described on line 247-8" publication bias that was evaluated using a funnel plot when 10 or more studies were in the same comparison". This is in line with the Cochranne Hadbook recommendations (https://handbook-5-1.cochrane.org/chapter_10/10_4_3_1_recommendations_on_testing_for_funnel_plot_asymmetry.htm). As observed in the comparisons made, there were no more than 10 clinical trials combined, so we did not perform a funnel plot, avoiding a mistaken interpretation of the results.

---

## [Decision Letter · Decision Letter 2]

27 Feb 2023

PONE-D-22-31577R2Acute effect of low-load resistance exercise with blood flow restriction on oxidative stress biomarkers: A systematic review and meta-analysisPLOS ONE

Dear Dr. Joao Vitor Ferlito,

Thank you for submitting your manuscript to PLOS ONE. After careful consideration, we feel that it has merit but does not fully meet PLOS ONE’s publication criteria as it currently stands. Therefore, we invite you to submit a revised version of the manuscript that addresses the points raised during the review process.

ACADEMIC EDITOR: Dear authors,

While the comments of reviewer 1 were already addressed, there are still few comments of reviewer 2 to improve your work. Please address them and retur the manuscript.

We believe that after this round, the manuscript has conditions to be accepted.

Thank you

We look forward to receiving your revised manuscript.

Kind regards,

Rafael Franco Soares Oliveira

Academic Editor

PLOS ONE

Journal Requirements:

Additional Editor Comments (if provided):

Dear authors,

While the comments of reviewer 1 were already addressed, there are still few comments of reviewer 2 to improve your work. Please address them and retur the manuscript.

We believe that after this round, the manuscript has conditions to be accepted.

Thank you

Reviewers' comments:

Reviewer's Responses to Questions

**Comments to the Author**

1. If the authors have adequately addressed your comments raised in a previous round of review and you feel that this manuscript is now acceptable for publication, you may indicate that here to bypass the “Comments to the Author” section, enter your conflict of interest statement in the “Confidential to Editor” section, and submit your "Accept" recommendation.

Reviewer #2: All comments have been addressed

2. Is the manuscript technically sound, and do the data support the conclusions?

Reviewer #2: Yes

3. Has the statistical analysis been performed appropriately and rigorously? 

Reviewer #2: Yes

4. Have the authors made all data underlying the findings in their manuscript fully available?

Reviewer #2: Yes

5. Is the manuscript presented in an intelligible fashion and written in standard English?

Reviewer #2: Yes

6. Review Comments to the Author

Reviewer #2: Dear Authors,

Thank you for considering my suggestions and incorporating them into the manuscript, which globally improved, congratulations. There are some details that still require special attention, namely the citation numbers and references format.

Below suggestions related to this last version, with line indication:

Table 1 – It is believed to be “de Lima” and not “Lima” in ref 45. Same in table 2. Please revise.

Table 1 & 2 and text manuscript. Ref 39 is not “Ramis”, should be corrected in tables and manuscript text or in references. This is very important because can determine errors throughout the document.

562 – Please consider not starting the phrase with the authors citation.

694-987 – Please carefully analyze and correct the references format considering the journal guidelines.

Please consider improving all figures quality.

Please revise the manuscript considering English details improvment.

7. PLOS authors have the option to publish the peer review history of their article (what does this mean?). If published, this will include your full peer review and any attached files.

Reviewer #2: No

---

## [Author Response · Author response to Decision Letter 2]

27 Feb 2023

Thank you for taking the time to make this manuscript the best it can be. We feel that your comments have helped shape the manuscript for the better. We hope you feel the paper is suitable for publication. 

Reviewer #2

Table 1 – It is believed to be “de Lima” and not “Lima” in ref 45. Same in table 2. Please revise.

Authors response: Thanks for the suggestion, we've adjusted the table as requested.

Table 1 & 2 and text manuscript. Ref 39 is not “Ramis”, should be corrected in tables and manuscript text or in references. This is very important because can determine errors throughout the document.

Authors response: Thanks for the suggestion, we've adjusted the document as requested.

562 – Please consider not starting the phrase with the authors citation.

Authors response: Thanks for the suggestion, we've adjusted the sentence. It now reads: 

“For example, there is evidence (62) demonstrating that low doses and short-term ROS exposure stimulate mTORC1 while high concentrations or long-term exposure ROS inhibit mTORC1 activity.”

694-987 – Please carefully analyze and correct the references format considering the journal guidelines.

Authors response: We have adjusted the manuscript to remove the “[Internet]” that was present that may have been the issue the reviewer pointed out. Otherwise, the manuscript is in Vancouver style according to our reference manager, Paperpile. According to PLOS: PLOS uses the reference style outlined by the International Committee of Medical Journal Editors (ICMJE), also referred to as the “Vancouver” style.

Please consider improving all figures quality.

Authors response: We are unsure what you are requesting as there is no direction provided by the reviewer. The images provided are from a computer program that is used to produce images for these types of analyses. We have already adjusted the figures in PR1 that met the satisfaction of the other reviewers (we redefined terms and standardized nomenclature used within these figures). If this is not adequate to your expectations, please specify what exactly you are requesting so we can make the changes. 

Please revise the manuscript considering English details improvment.

Authors response: We have gone through and adjusted the manuscript once more and the new edits are in red throughout the text. Thank you. Of note: the co-first author is an English speaker fluent in academic writing and has been published elsewhere. If these changes are still not suitable, we are unsure exactly of what this reviewer is requesting given the vague feedback. Please specify exactly what is not up to publication standards in this journal so we can make the appropriate adjustments. Thank you!

---

## [Editor Report · Decision Letter 3]

1 Mar 2023

PONE-D-22-31577R3Acute effect of low-load resistance exercise with blood flow restriction on oxidative stress biomarkers: A systematic review and meta-analysisPLOS ONE

Dear Dr. Joao Vitor Ferlito,

Thank you for submitting your manuscript to PLOS ONE. After careful consideration, we feel that it has merit but does not fully meet PLOS ONE’s publication criteria as it currently stands. Therefore, we invite you to submit a revised version of the manuscript that addresses the points raised during the review process.

ACADEMIC EDITOR:Dear authors,

All comments made by reviewers were addressed and the paper can be accepted. However, I found that flow chart is not following PRISMA 2020 guidelines. In additon, authors used "Data management and study selection process" plus ""Data Extraction" which are not present in PRISMA 2020. Please revise those issues and resubmit. I hope in the next stage, I can accept your work.

Best regards

We look forward to receiving your revised manuscript.

Kind regards,

Rafael Franco Soares Oliveira

Academic Editor

PLOS ONE

Journal Requirements:

Additional Editor Comments (if provided):

Dear authors,

All comments made by reviewers were addressed and the paper can be accepted. However, I found that flow chart is not following PRISMA 2020 guidelines. In additon, authors used "Data management and study selection process" plus ""Data Extraction" which are not present in PRISMA 2020. Please revise those issues and resubmit. I hope in the next stage, I can accept your work.

Best regards
---

## [Author Response · Author response to Decision Letter 3]

2 Mar 2023

All comments made by reviewers were addressed and the paper can be accepted. However, I found that flow chart is not following PRISMA 2020 guidelines. In additon, authors used "Data management and study selection process" plus ""Data Extraction" which are not present in PRISMA 2020. Please revise those issues and resubmit. I hope in the next stage, I can accept your work.

Authors Reply: Editor – we appreciate your attention to detail and apologize for our formatting and labeling issues. We have addressed the issues you have brought up in our manuscript and we hope that the paper is now suitable for formal acceptance. We relabeled the sections “Selection process” (ln 168) and “Data collection process” (ln 179). We have also reformatted Figure 1 as per PRISMA 2020 guidelines. Last, we made sure to address the “de” in front of Lima in 2 of our tables. Thank you again for the opportunity to publish in your journal.

---

## [Editor Report · Decision Letter 4]

6 Mar 2023

Acute effect of low-load resistance exercise with blood flow restriction on oxidative stress biomarkers: A systematic review and meta-analysis

PONE-D-22-31577R4

Dear Dr. Joao Vitor Ferlito,

We’re pleased to inform you that your manuscript has been judged scientifically suitable for publication and will be formally accepted for publication once it meets all outstanding technical requirements.

Kind regards,

Rafael Franco Soares Oliveira

Academic Editor

PLOS ONE

Additional Editor Comments (optional):

Dear authors,

Congratulations! Your work is now ready for publication!

Thank you

Best regards
---

## [Editor Report · Acceptance letter]

9 Mar 2023

PONE-D-22-31577R4 

Acute effect of low-load resistance exercise with blood flow restriction on oxidative stress biomarkers: A systematic review and meta-analysis 

Dear Dr. Ferlito:

I'm pleased to inform you that your manuscript has been deemed suitable for publication in PLOS ONE. Congratulations! Your manuscript is now with our production department. 

Kind regards, 

on behalf of

Prof Rafael Franco Soares Oliveira 

Academic Editor

PLOS ONE